# HT-Net: Hierarchical Transformer based Operator Learning Model for Multiscale PDEs

## Abstract

Complex nonlinear interplays of multiple scales give rise to many interesting physical phenomena and pose significant difficulties for the computer simulation of multiscale PDE models in areas such as reservoir simulation, high-frequency scattering, and turbulence modeling. In this paper, we introduce a hierarchical transformer (HT) scheme to efficiently learn the solution operator for multiscale PDEs. We construct a hierarchical architecture with a scale-adaptive interaction range, such that the features can be computed in a nested manner and with a controllable linear cost. Self-attentions over a hierarchy of levels can be used to encode and decode the multiscale solution space across all scales. In addition, we adopt an empirical $H^1$ loss function to counteract the spectral bias of the neural network approximation for multiscale functions. In the numerical experiments, we demonstrate the superior performance of the HT scheme compared with state-of-the-art (SOTA) methods for representative multiscale problems.

## 1 Introduction

Partial differential equation (PDE) models with multiple temporal/spatial scales are ubiquitous in physics, engineering, and other disciplines. They are of tremendous importance in making predictions for challenging practical problems such as reservoir modeling, high-frequency scattering, and atmosphere circulation, to name a few. The complex nonlinear interplays of characteristic scales cause major difficulties in the computer simulation of multiscale PDEs. While the resolution of all characteristic scales is prohibitively expensive, sophisticated multiscale methods have been developed to efficiently and accurately solve the multiscale PDEs by incorporating microscopic information. However, most of them are designed for problems with fixed input parameters.

Recently, several novel methods such as Fourier neural operator (FNO) (Li et al., 2021), Galerkin transformer (GT) (Cao, 2021) and deep operator network (DeepONet) (Lu et al., 2021) are developed to directly learn the operator (mapping) between infinite dimensional spaces for PDE problems, by taking advantages of the enhanced expressibility of deep neural networks and advanced architectures such as feature embedding, channel mixing, and self-attentions. Such methods can deal with an ensemble of input parameters and have great potential for the efficient forward and inverse solvers of PDE problems. However, for multiscale problems, most existing operator learning schemes essentially capture the smooth part of the solution space, and how to resolve the intrinsic multiscale features remains to be a major challenge.

In this paper, we design a hierarchical transformer based operator learning method, so that the accurate, efficient, and robust computer simulation of multiscale PDE problems with an ensemble of input parameters becomes feasible. Our main contribution can be summarized in the following,

- we develop a novel transformer architecture that allows the decomposition of the input-output mapping to a hierarchy of levels, the features can be updated in a nested manner based on the hierarchical local aggregation of self-attentions with linear computational cost;
- we adopt the empirical $H^1$ loss which avoids the spectral bias and enhances the ability to capture the oscillatory features of the multiscale solution space;
- the resulting scheme has significantly better accuracy and generalization properties for multiscale input parameters, compared with state-of-the-art (SOTA) models.

## 2 BACKGROUND AND RELATED WORK

We briefly introduce multiscale PDEs in Section § 2.1, then summarize relevant multiscale numerical methods in § 2.2, and neural solvers, in particular neural operators in § 2.3.

### 2.1 MULTISCALE PDEs

Multiscale PDEs, in a narrower sense, refer to PDEs with rapidly varying coefficients, which may arise from a wide range of applications in heterogeneous and random media. In a broader sense, they may form a hierarchy of models at different scales by a systematic derivation, starting from fundamental laws of physics such as quantum mechanics (E & Engquist, 2003). Some outstanding multiscale PDE models may include:

Multiscale elliptic equations is an important class of prototypical examples, such as the following second-order elliptic equation of divergence form,

$$
\begin{aligned}
-\nabla \cdot (a(x)\nabla u(x)) &= f(x) & x \in D \\
u(x) &= 0 & x \in \partial D
\end{aligned}
\tag{2.1}
$$

where $0 < a_{\min} \le a(x) \le a_{\max}, \forall x \in D$, and the forcing term $f \in H^{-1}(D; \mathbb{R})$. The coefficient to solution map is $\mathcal{S} : L^\infty(D; \mathbb{R}_+) \to H_0^1(D; \mathbb{R})$, such that $u = \mathcal{S}(a)$. In Li et al. (2021), smooth coefficient $a(x)$ is considered and $\mathcal{S}$ can be well resolved by the FNO parameterization. The setup $a(x) \in L^\infty$ allows rough coefficients with fast oscillation (e.g. $a(x) = a(x/\varepsilon)$ with $\varepsilon \ll 1$), high contrast ratio with $a_{\max}/a_{\min} \gg 1$, and even a continuum of non-separable scales. The rough coefficient case is much harder from both scientific computing (Branets et al., 2009) and operator learning perspectives.

Navier-Stokes equation models the flow of incompressible fluids, which becomes turbulent due to the simultaneous interaction of a wide range of temporal and spatial scales of motion.

Helmholtz equation models time-harmonic acoustic waves. Its numerical solution exhibits severe difficulties in the high wave number regime due to the interaction of high-frequency waves and numerical mesh.

### 2.2 MULTISCALE SOLVERS FOR MULTISCALE PDEs

For multiscale PDEs, the computational cost of classical numerical methods, such as finite element methods, finite difference methods, etc., usually scales proportionally to $1/\varepsilon \gg 1$. Multiscale solvers have been developed such that their computational costs are independent of $\varepsilon$ by incorporating microscopic information.

**Asymptotic and numerical homogenization** Asymptotic homogenization (Bensoussan et al., 1978) is an elegant analytical approach for multiscale PDEs with scale separation, e.g., $a(x) = a(x/\varepsilon)$ with $\varepsilon \ll 1$ in equation 2.1. For general multiscale PDEs with possibly a continuum of scales, numerical homogenization (Engquist & Souganidis, 2008) offers an effective numerical approach that aims to identify low dimensional approximation spaces such that the approximation bases are adapted to the corresponding multiscale operator, and can be efficiently constructed (e.g. with localized bases). See Appendix A for details.

**Multilevel and multiresolution methods** Multilevel/multigrid methods (Hackbusch, 1985; Xu & Zikatanov, 2017), and multiresolution methods such as wavelets (Brewster & Beylkin, 1995; Beylkin & Coult, 1998) have been successfully applied to PDEs. However, the convergence of those methods for multiscale problems can be severely affected by the regularity of coefficients (Branets et al., 2009). Recently, the introduction of gamblets (Owhadi, 2017) can be seen as a multilevel extension of numerical homogenization, and it opens an avenue to automatically discover scalable multilevel algorithms and operator-adapted wavelets for linear PDEs with rough coefficients. See also Appendix A.2.

**Low-rank decomposition based methods** It is well-known that the elliptic Green's function has low-rank approximation (Bebendorf, 2005), which lays the theoretical foundation of the (near-)linear complexity methods such as fast multipole method (Greengard & Rokhlin, 1987; Ying et al., 2004), hierarchical matrices ($\mathcal{H}$ and $\mathcal{H}^2$ matrices) (Hackbusch et al., 2002; Bebendorf, 2008), and hierarchical interpolative factorization (Ho & Ying, 2016). See Appendix B for the connection with our method presented in this paper.

**Tensor numerical methods** Analytical approaches such as periodic unfolding (Cioranescu et al., 2008) suggest that a low-dimensional multiscale PDE can be transformed into a high-dimensional PDE. Therefore, tensor numerical methods, e.g., the sparse tensor finite element method (Harbrecht & Schwab, 2011) and the quantic tensor train (QTT) method (Kazeev et al., 2022), provide efficient numerical procedures to find the low-rank tensor representation of multiscale solutions.

## 2.3 NEURAL OPERATOR FOR PDES

**Neural PDE solvers and Spectral Bais** Various neural solvers have been proposed in E et al. (2017); Sirignano & Spiliopoulos (2018); Raissi et al. (2019) to solve PDEs with fixed parameters. The difficulty to solve multiscale PDEs has been exhibited by the so-called spectral bias or frequency principle (Rahaman et al., 2019; Ronen et al., 2019), which shows that DNN-based algorithms are often inefficient to learn high-frequency components of multiscale functions. A series of algorithms have been developed to overcome the high-frequency curse of DNNs and to solve multiscale PDEs (Cai et al., 2020; Wang et al., 2021).

**Operator learning for PDEs** DNN algorithms demonstrate more potential to learn the input-output mapping of parametric PDEs. Finite-dimensional operator learning methods such as Zhu & Zabaras (2018); Fan et al. (2019a;b); Khoo et al. (2020) can be applied to problems with fixed discretization. In particular, Fan et al. (2019a;b) combines $\mathcal{H}$ or $\mathcal{H}^2$ matrices linear operations with nonlinear activation functions, though the nonlinear geometrical interaction/aggregation is absent, which limits the expressivity of the resulting neural operator. Infinite-dimensional operator learning methods (Li et al., 2021; Gupta et al., 2021) aim to learn the mapping between infinite-dimensional Banach spaces, and the convolutions in the neural operator construction are parametrized by Fourier or wavelet transform. Nevertheless, those methods do not always work even for multiscale PDEs with fixed parameters. On the other hand, while the universal approximation can be rigorously proved for FNO operators (Kovachki et al., 2021), "extra smoothness" is required to achieve a meaningful decay rate, which either is absent or gives rise to large constants for multiscale PDEs. This motivates us to construct a new architecture for multiscale operator learning.

**Efficient Attention** The vanilla multi-head self-attention in Vaswani et al. (2017) scales quadratically with the number of tokens, which is prohibitive for high-resolution problems. Many efficient attention methods are proposed using the kernel trick (low-rank projection) (Choromanski et al., 2020; Wang et al., 2020; Peng et al., 2021; Nguyen et al., 2021) to reduce the computation cost arising in the dense matrix operation. In the context of operator learning, Galerkin transformer (Cao, 2021) proposed to remove the softmax normalization in self-attention and introduced a linearized self-attention variant with Petrov-Galerkin projection normalization. Furthermore, hierarchical transformers using local window aggregation are proposed in Liu et al. (2021b); Zhang et al. (2022) for NLP and vision applications.

## 3  METHODS

We introduce the hierarchical attention model in this section, motivated by the hierarchical matrix, in particular the $\mathscr{H}^2$ matrix variant. We follow the setup in Li et al. (2021); Lu et al. (2021) to approximate the operator $\mathcal{S} : \boldsymbol{a} \mapsto \boldsymbol{u} := \mathcal{S}(\boldsymbol{a})$, with the input $\boldsymbol{a} \in \mathcal{A}$ drawn from a distribution $\mu$ and the output $\boldsymbol{u} \in \mathcal{U}$, where $\mathcal{A}$ and $\mathcal{U}$ are infinite dimensional Banach spaces respectively. We aim to learn the operator $\mathcal{S}$ from a finite collection of finitely observed input-output pairs through a parametric map $\mathcal{N} : \mathcal{A} \times \Theta \to \mathcal{U}$ and a loss functional $\mathcal{L} : \mathcal{U} \times \mathcal{U} \to \mathbb{R}$, such that the optimal parameter $\theta^* = \arg\min_{\theta \in \Theta} \mathbb{E}_{\boldsymbol{a} \sim \mu} \left[ \mathcal{L} \left( \mathcal{N}(\boldsymbol{a}, \theta), S(\boldsymbol{a}) \right) \right]$. The input $\boldsymbol{a}$ can then be mapped to features $\boldsymbol{f}$ through patch embedding (Dosovitskiy et al., 2020).

**Hierarchical Discretization**  We introduce the hierarchical discretization of the spatial domain $D$, which can be used directly for time-independent PDEs. Time-dependent PDEs can be treated by taking time-sliced data as feature channels. Let $\mathcal{I}^{(r)}$ be the finest level index set, such that each index $i = (i_1, \ldots, i_r) \in \mathcal{I}^{(r)}$ denotes the finest level spatial objects such as image pixels, finite difference points, etc. For any $i = (i_1, \ldots, i_r) \in \mathcal{I}^{(r)}$, and $1 \leq m \leq r$, $i^{(m)} = (i_1, \ldots, i_m)$ represents $i$'s $m$-th level parent node which is the aggregate of finer level objects. $\mathcal{I}^{(m)} := \{ i^{(m)} : i \in \mathcal{I}^{(r)} \}$ is the $m$-th level index set. We can similarly define the natural parent-child relationship between coarser and finer nodes, which induces the index tree $\mathcal{I}$ together with the index sets $\mathcal{I}^{(m)} (1 \leq m \leq r)$. Each index $i \in \mathcal{I}^{(m)}$ corresponds to a token, which can be characterized e.g. by its position $\boldsymbol{x}_i^{(m)}$ and a feature vector $\boldsymbol{f}_i^{(m)}$ with number of channels $\mathcal{C}^{(m)}$. In the following, we will restrict our presentation to the quadtree setting as illustrated in Fig 3.1.

**Reduce Operation**  The reduce operation defines the map from finer-level features to coarser-level features. For $i \in \mathcal{I}^{(m)}$, we denote $i^{(m,m+1)}$ as the set of $(m+1)$-th level child nodes of $i$. In the quadtree setting, $i^{(m,m+1)} = \{ (i,0), (i,1), (i,2), (i,3) \}^1$. The reduce map can be abstractly defined as $\boldsymbol{f}_i^{(m),t} = \mathcal{R}^{(m)}(\{ \boldsymbol{f}_j^{(m+1),t} \}_{j \in i^{(m,m+1)}})$, which maps the $(m+1)$-th level features with indices in $i^{(m,m+1)}$ to the $m$-th level feature with index $i^{(m)}$. For simplicity, we use a simple linear layer for the operator $\mathcal{R}^{(m)}$ in our current implementation, namely, $\boldsymbol{f}_i^{(m),t} = \boldsymbol{R}_0^{(m)} \boldsymbol{f}_{(i,0)}^{(m+1),t} + \boldsymbol{R}_1^{(m)} \boldsymbol{f}_{(i,1)}^{(m+1),t} + \boldsymbol{R}_2^{(m)} \boldsymbol{f}_{(i,2)}^{(m+1),t} + \boldsymbol{R}_3^{(m)} \boldsymbol{f}_{(i,3)}^{(m+1),t}$, where $\boldsymbol{R}_0^{(m)}, \boldsymbol{R}_1^{(m)}, \boldsymbol{R}_2^{(m)}, \boldsymbol{R}_3^{(m)} \in \mathbb{R}^{\mathcal{C}^{(m-1)} \times \mathcal{C}^{(m)}}$ are parametrized by linear layers. From the perspective of $\mathcal{H}$ matrices, those parametrized matrices $\boldsymbol{R}_j^{(m)}$ embed low-rank approximation to the attention kernel. Please refer to Appendix B for more details.

**Multilevel Local Aggregation**  The finest level features $\boldsymbol{f}_i^{(r),t} \in \mathbb{R}^{\mathcal{C}^{(r)}}$ at the evolution step $t$, with index $i \in \mathcal{I}^{(r)}$, can be updated by the following token aggregation formula through vanilla attention,

$$\text{atten} : \boldsymbol{f}_i^{(r),t+1} = \sum_{j=1}^{N^{(r)}} \mathcal{G}(\boldsymbol{f}_i^{(r),t}, \boldsymbol{f}_j^{(r),t}) v(\boldsymbol{f}_j^{(r),t}), \text{ for } i \in \mathcal{I}^{(r)}, \tag{3.1}$$

with $N^{(r)} := |\mathcal{I}^{(r)}|$. For simplicity, we ignore the softmax normalizing factor and only consider single-head attention, and we assume that the interaction kernel $\mathcal{G}$ is of the form $\mathcal{G}(\boldsymbol{f}_i, \boldsymbol{f}_j) = \exp(\boldsymbol{W}^Q \boldsymbol{f}_i \cdot \boldsymbol{W}^K \boldsymbol{f}_j) = \exp(\boldsymbol{q}_i \cdot \boldsymbol{k}_j)$, where $\boldsymbol{q}_i := \boldsymbol{W}^Q \boldsymbol{f}_i$, $\boldsymbol{k}_i := \boldsymbol{W}^K \boldsymbol{f}_i$, $\boldsymbol{v}_i := v(\boldsymbol{f}_i) = \boldsymbol{W}^V \boldsymbol{f}_i$, and $\boldsymbol{W}^Q, \boldsymbol{W}^K, \boldsymbol{W}^V \in \mathbb{R}^{\mathcal{C}^{(r)} \times \mathcal{C}^{(r)}}$ are learnable matrices.

Instead of computing equation 3.1 explicitly with $\mathcal{O}(N^2)$ cost, we propose a self-attention based local aggregation scheme, inspired by the $\mathscr{H}^2$ matrices (Hackbusch, 2015), see also Appendix B. The local aggregation at the $r$-th level writes,

---

$^1 (i, j)$ is understood as the concatenation of entries of $i$ and the scalar $0 \leq j \leq 3$.

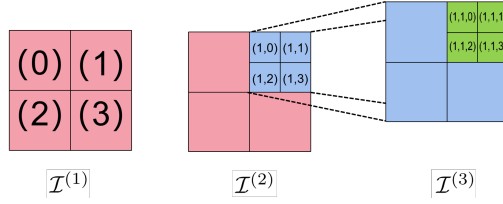

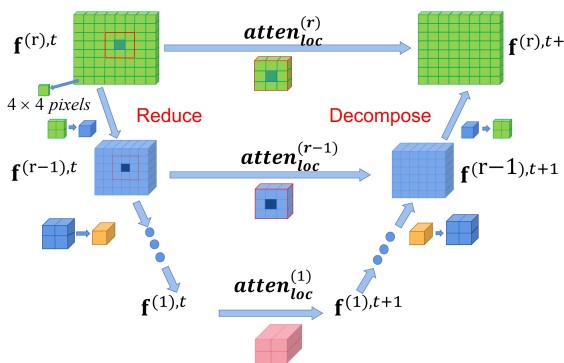

Figure 3.1: Hierarchical discretization and index tree. In this example, the 2D unit square is hierarchically discretized into three levels which are indexed by $\mathcal{I}^{(1)}, \mathcal{I}^{(2)}$ and $\mathcal{I}^{(3)}$, respectively. For example, we denote by $(1)^{(1,2)} = \{(1,0), (1,1), (1,2), (1,3)\}$ the set of the second level child nodes of the node $(1)$.

Figure 3.2: One V-cycle of the feature update.

$$\text{atten}_{\text{loc}}^{(r)} : \boldsymbol{f}_i^{(r),t+1} = \sum_{j \in \mathcal{N}^{(r)}(i)} \exp(\boldsymbol{q}_j^{(r),t} \cdot \boldsymbol{k}_j^{(r),t}) \boldsymbol{v}_j^{(r),t}, \text{ for } i \in \mathcal{I}^{(r)}, \tag{3.2}$$

where $\mathcal{N}^{(r)}(i)$ is the set of the $r$-th level neighbors of $i \in \mathcal{I}^{(r)}$, $\boldsymbol{q}_i^{(r),t} := \hat{\boldsymbol{W}}^Q \boldsymbol{f}_i^{(r),t}$, $\boldsymbol{k}_i^{(r),t} := \hat{\boldsymbol{W}}^K \boldsymbol{f}_i^{(r),t}$, $\boldsymbol{v}_i^{(r),t} := \hat{\boldsymbol{W}}^V \boldsymbol{f}_i^{(r),t}$ with learnable matrices $\hat{\boldsymbol{W}}^Q, \hat{\boldsymbol{W}}^K, \hat{\boldsymbol{W}}^V \in \mathbb{R}^{\mathcal{C}^{(r)} \times \mathcal{C}^{(r)}}$. Then for $m = r-1, ..., 1$, we calculate the local attention in each level with nested $\boldsymbol{q}_i^{(m),t}, \boldsymbol{k}_j^{(m),t}, \boldsymbol{v}_j^{(m),t}$,

$$\text{atten}_{\text{loc}}^{(m)} : \boldsymbol{f}_i^{(m),t+1} = \sum_{j \in \mathcal{N}^{(m)}(i)} \exp(\boldsymbol{q}_i^{(m),t} \cdot \boldsymbol{k}_j^{(m),t}) \boldsymbol{v}_j^{(m),t}, \text{ for } i \in \mathcal{I}^{(m)} \tag{3.3}$$

where $\boldsymbol{q}_i^{(m),t} = \mathcal{R}^{(m)}(\{\boldsymbol{q}_j^{(m+1),t}\}_{j \in i^{(m,m+1)}})$, $\boldsymbol{k}_i^{(m),t} = \mathcal{R}^{(m)}(\{\boldsymbol{k}_j^{(m+1),t}\}_{j \in i^{(m,m+1)}})$, $\boldsymbol{v}_i^{(m),t} = \mathcal{R}^{(m)}(\{\boldsymbol{v}_j^{(m+1),t}\}_{j \in i^{(m,m+1)}})$, $\mathcal{N}^{(m)}(i)$ is the set of the $m$-th level neighbors of $i \in \mathcal{I}^{(m)}$.

**Decompose Operation** To mix the multilevel features $\boldsymbol{f}_i^{(m),t+1}, m = 1, ..., r$, we propose a decompose operation that reverses the reduce operation from level $1$ to level $r-1$. The decompose operator $\mathcal{D}^{(m)} : \boldsymbol{f}_i^{(m),t+1} \mapsto \{\boldsymbol{f}_j^{(m+1),t+\frac{1}{2}}\}_{j \in i^{(m,m+1)}}$, maps the $m$-th level feature with index $i$ to $(m+1)$-th level features associated to its child set $i^{(m,m+1)}$. $\boldsymbol{f}_i^{(m+1),t+\frac{1}{2}}$ is further aggregated to $\boldsymbol{f}_i^{(m+1),t+1}$ such that $\boldsymbol{f}_i^{(m+1),t+1} + = \boldsymbol{f}_i^{(m+1),t+\frac{1}{2}}$ for $i \in \mathcal{I}^{(m+1)}$. In the current implementation, we use a simple linear layer such that $\boldsymbol{f}_{(i,s)}^{(m+1),t+\frac{1}{2}} = \boldsymbol{D}_s^{(m),T} \boldsymbol{f}_i^{(m),t+1}$, for $s = 0, 1, 2, 3$, with parameter matrices $\boldsymbol{D}_s^{(m)} \in \mathbb{R}^{\mathcal{C}^{(m)} \times \mathcal{C}^{(m+1)}}$.

**Remark 1** *In our current implementation, the operators $\mathcal{R}^{(m)}$ and $\mathcal{D}^{(m)}$ only consist of linear layers. In general, these operators may contain nonlinear activation functions.*

**Remark 2** *The nested learnable operators $\mathcal{R}^{(m)}$ and $\mathcal{D}^{(m)}$ induce the channel mixing and a structured parameterization of $\boldsymbol{W}^Q, \boldsymbol{W}^V, \boldsymbol{W}^K$ matrix for the coarse level tokens. See Appendix B for details.*

We summarize the hierarchical attention algorithm in the following.

---

**Algorithm 1** One V-cycle of Hierarchical Attention

---

**Input**: $\mathcal{I}^{(r)}$, $\boldsymbol{f}_i^{(r),t}$ for $i \in \mathcal{I}^{(r)}$.

**STEP 0:** Get the $\boldsymbol{q}_i^{(r),t}$, $\boldsymbol{k}_i^{(r),t}$, $\boldsymbol{v}_i^{(r),t}$ for $i \in \mathcal{I}^{(r)}$.

**STEP 1:** For $m = r - 1, \cdots, 1$, **Do** the reduce operations $\boldsymbol{q}_i^{(m),t} = \mathcal{R}^{(m)}(\{\boldsymbol{q}_j^{(m+1),t}\}_{j \in i(m,m+1)})$ and also for $\boldsymbol{k}_i^{(m),t}$ and $\boldsymbol{v}_i^{(m),t}$, for any $i \in \mathcal{I}^{(m)}$.

**STEP 2:** For $m = r, \cdots, 1$, **Do** the local aggregation by equation 3.2 and equation 3.3 to get $\boldsymbol{f}_i^{(m),t+1}$, $m = 1, ..., r$ for any $i \in \mathcal{I}^{(m)}$.

**STEP 3:** For $m = 1, \cdots, r - 1$, **Do** the decompose operations $\{\boldsymbol{f}_j^{(m+1),t+\frac{1}{2}}\}_{j \in i(m,m+1)} = \mathcal{D}^{(m)}(\boldsymbol{f}_i^{(m),t+1})$, for any $i \in \mathcal{I}^{(m)}$; then $\boldsymbol{f}_i^{(m+1),t+1} += \boldsymbol{f}_i^{(m+1),t+\frac{1}{2}}$, for any $i \in \mathcal{I}^{(m+1)}$

**Output**: $\boldsymbol{f}_i^{(r),t+1}$ for any $i \in \mathcal{I}^{(r)}$.

---

**Proposition 3.1 (Complexity of Algorithm 1)** *The reduce operation, multilevel aggregation, and decomposition operation together form a V-cycle for the update of features, as illustrated in Figure 3.2. The cost of one V-cycle is $O(N)$ if $\mathcal{I}$ is a quadtree as implemented in the paper. See Appendix C for the proof.*

**Remark 3 (Technical contribution)** *In Liu et al. (2021b); Zhang et al. (2022), spatial resolutions are sequentially reduced, and attentions are performed at each level separately. However, attentions are only used for a fixed level, and there is no multilevel attention-based aggregation. The coarse latent variables are used for classification and generation tasks, and fine-scale information may get lost in the reduce(coarsening) process. While in the HT-Net architecture, we are motivated by multiscale numerical methods such as numerical homogenization and hierarchical matrix method. Attention-based local aggregations are constructed at each level, and features from all levels are summed up to form the updated fine-scale features, which enables the recovery of the fine details with linear cost.*

**Decoder** The decoder maps the features $\boldsymbol{f}$ (at the last update step) to the solution $\boldsymbol{u}$. The decoder is often chosen with prior knowledge of the PDE. A simple feedforward neural network (FFN) is used in Lu et al. (2021) to learn a basis set as the decoder. A good decoder can also be learned from the data using SVD (Bhattacharya et al., 2021). In this paper, we use the spectral convolution layers in Li et al. (2021).

**Loss functions** Loss functions are crucial for efficient training and robust generalization of neural network models. For multiscale problems considered in this paper, we prefer to use $H^1$ loss instead of the usual $L^2$ loss function, as it puts more "weights" on high-frequency components. Yu et al. (2022) adopted Sobolev norm based loss function for function approximation, where $H^1$ loss is defined for the neural network function itself. On the contrary, we define the $H^1$ loss on the target solution space, which measures the distance between the prediction $\hat{\boldsymbol{u}} := \mathcal{N}(\boldsymbol{a})$ and the ground truth $\boldsymbol{u}$. We refer the readers to Adams & Fournier (2003) for the definition of Sobolev spaces and $H^1$ norm, and to the next section for the implementation.

## 4 EXPERIMENTS

We demonstrate the effectiveness of the hierarchical transformer model (HT-Net) on multiscale operator learning. All our examples are in 2D. We first study the ability of the model on a multiscale elliptic equation with two-phase coefficients which is a widely used benchmark problem for operator learning (Li et al., 2021). As the "multiscaleness" of the coefficients such as the oscillation, contrast, and roughness of the two-phase interface can be fine-tuned, we showcase the accuracy and the robustness of the HT-NET model for both smooth and rough coefficients. We also show that HT-NET is able to provide better generalization errors for out-of-distribution input parameters. We also test the Navier-Stokes equation with a large Reynolds number as an example with non-linearity and time dependence.

## 4.1 SETUP

In our experiments, the spatial domain is $D := [0,1]^2$ and is discretized uniformly with $h = 1/n$. Let the uniform grid be $\mathsf{G}^2 := \{(x_i, x_j) = (ih, jh) \mid i,j = 0, ..., n-1\}$. $\{(\boldsymbol{a}_j, \boldsymbol{u}_j)\}_{j=1}^N$ are functions pairs such that $\boldsymbol{u}_j = \mathcal{S}(\boldsymbol{a}_j)$, and $\boldsymbol{a}_j$ is drown from some probability measure $\mu$. The actual data pairs for training and testing are pointwise evaluations of $\boldsymbol{a}_j$ and $\boldsymbol{u}_j$ on the grid $\mathsf{G}^2$, denoted by $a_j$ and $u_j$, respectively. The comparison with all the baselines is consistent with (most time better than) references. The hierarchical index tree $\mathcal{I}$ can be generated by the corresponding quadtree representation of the nodes with depth $r$, such that the finest level objects are pixels or patches aggregated by pixels. See Appendix D for more details.

**Empirical $H^1$ loss function** Empirical $L^2$ loss function is defined as $\mathcal{L}^L(\{(\boldsymbol{a}_j, \boldsymbol{u}_j)\}_{j=1}^N; \theta) := \frac{1}{N}\sum_{i=1}^N \|\boldsymbol{u}_j - \mathcal{N}(\boldsymbol{a}_j; \theta)\|_{l^2}/\|\boldsymbol{u}_j\|_{l^2}$, where $\|\cdot\|_{l^2}$ is the canonical $l^2$ vector norm. For any $\xi \in \mathbb{Z}_n^2 := \{\xi \in \mathbb{Z}^2 \mid -n/2 + 1 \leqslant \xi_j \leqslant n/2, j = 1, 2\}$, the normalized discrete Fourier transform (DFT) coefficients of $f$ writes $\mathcal{F}(f)(\xi) := \frac{1}{\sqrt{n}}\sum_{x \in \mathsf{G}^2} f(x)e^{-2i\pi x \cdot \xi}$. The empirical $H^1$ loss function is given by, $\mathcal{L}^H(\{(\boldsymbol{a}_j, \boldsymbol{u}_j)\}_{j=1}^N; \theta) := \frac{1}{N}\sum_{i=1}^N \|\boldsymbol{u}_j - \mathcal{N}(\boldsymbol{a}_j; \theta)\|_h/\|\boldsymbol{u}_j\|_h$, where $\|u\|_h := \sqrt{\sum_{\xi \in \mathbb{Z}_n^2} |\xi|^2 (\mathcal{F}(u)(\xi))^2}$. $\mathcal{L}^H$ can be viewed as a weighted $\mathcal{L}^L$ loss with weights $|\xi|^2$, which forces the operator to capture high-frequency components in the solution. In this work, we use the equivalent frequency space representation of the discrete $H^1$ norm, and in general, discrete $H^1$ norm in real space can also be adopted.

## 4.2 MULTISCALE ELLIPTIC EQUATION

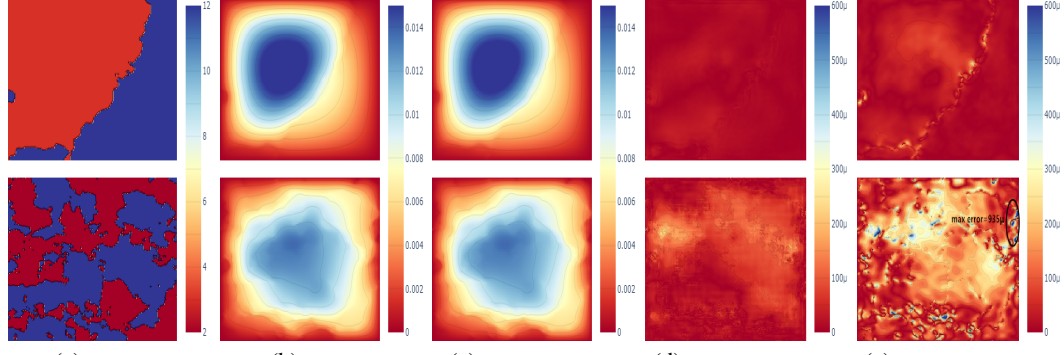

| (a) coefficient | (b) reference | (c) HT-Net prediction | (d) abs. error of HT-Net | (e) abs. error of FNO2D |

Figure 4.1: **Top:** (a) smooth coefficient in Li et al. (2021), with $a_{\max} = 12$, $a_{\min} = 3$ and $c = 9$, (b), reference solution, (c) HT-Net solution, (d) absolute error of HT-Net, (e) absolute error of FNO2D; **Bottom:** (a) rough coefficients with $a_{\max} = 12$, $a_{\min} = 2$ and $c = 20$, (b) reference solution, (c) HT-Net solution, (d) absolute error of HT-Net, (e) absolute error of FNO2D, The maximal absolute error in **Bottom:**(e) is around $900\mu = 9\mathrm{e}{-4}$.

We apply the HT-Net model to learn the coefficient to solution operator for equation 2.1. We use the two-phase coefficient model in Li et al. (2021), which is also benchmarked in Gupta et al. (2021); Cao (2021). In previous works, the coefficients do not oscillate much and the solutions look smooth. We change the parameters which control the smoothness and contrast of the coefficients, such that the solutions contain more roughness than the Darcy benchmark in Li et al. (2021), see Figure 4.1. We refer the readers to Appendix E for data generation details. We also include experiments for multiscale trigonometric coefficients with higher contrast. Detailed description on multiscale trigonometric coefficients is in Appendix F.1.

Table 1: Performance on multiscale elliptic equation. For the Darcy rough case, we run each experiment 3 times to calculate the mean and the standard deviation (after $\pm$) for relative $L^2$ errors ($\times 10^{-2}$) and relative $H^1$ errors ($\times 10^{-2}$). All experiments use a fixed train-val-test split setup, see Appendix D for details. SWIN is constructed by adding the encoder-decoder architeture of HT-Net to the original SWIN code, and is served as a baseline for multiscale vision transformers. FNO2D-48 and FNO2D-96 are adapted from FNO2D (by default, with 12 modes) with 48 modes and 96 modes respectively in order to learn high-frequency outputs. HT-Net outperforms other neural operators by a considerable margin for all three cases.

| | | Darcy smooth | | Darcy rough | | Multiscale trigonometric | |
|---|---|---|---|---|---|---|---|
| **Model** | Runtime (s) | $L^2$ | $H^1$ | $L^2$ | $H^1$ | $L^2$ | $H^1$ |
| FNO2D | 7.278 | 0.620 | 3.883 | 1.646 $\pm$0.021 | 11.955$\pm$0.088 | 1.794 | 12.605 |
| FNO2D-48 | 8.062 | 0.619 | 2.620 | 1.220 $\pm$0.018 | 5.138 $\pm$0.093 | 1.565 | 11.093 |
| FNO2D-96 | 10.969 | 0.575 | 2.437 | 1.216 $\pm$0.024 | 5.140 $\pm$0.281 | 1.518 | 10.106 |
| MWT | 19.715 | — | — | 1.138 $\pm$0.010 | 4.107 $\pm$0.008 | 1.021 | 7.245 |
| GT | 38.219 | 0.945 | 3.365 | 1.790 $\pm$0.012 | 6.269 $\pm$0.418 | 1.052 | 8.207 |
| SWIN | 41.417 | — | — | 1.622 $\pm$0.047 | 6.796 $\pm$0.359 | 1.489 | 13.385 |
| HT-NET | 33.375 | **0.291** | **0.815** | **0.571$\pm$0.001** | **1.371$\pm$0.001** | **0.603** | **2.633** |

— MWT (Gupta et al., 2021) only supports resolution with powers of two.

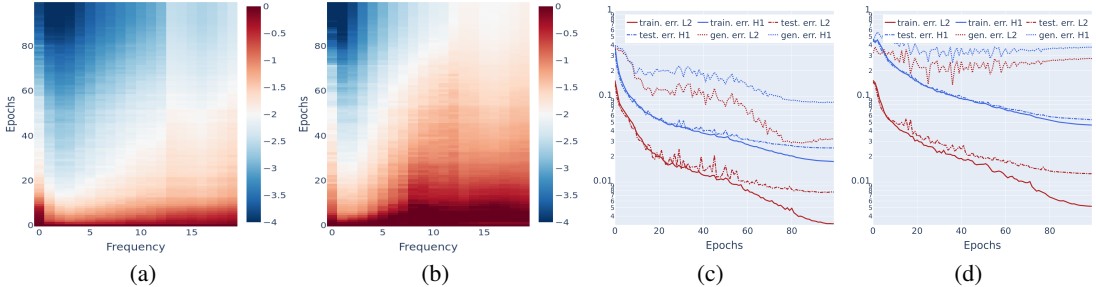

(a)             (b)             (c)             (d)

Figure 4.2: In (a) HT-Net trained with $H^1$ loss, and (b) HT-Net trained with $L^2$ loss, we show the evolution of errors with x-axis for frequency, y-axis for training epochs, and colorbar for the magnitude of $L^2$ error on each frequency in $\log_{10}$ scale, the error for each frequency is normalized frequency-wise by the error at epoch 0. The loss curves with training, testing, and generalization errors are shown in (c) for HT-Net trained with $H^1$ loss, and in (d) for HT-Net trained with $L^2$ loss.

**Spectral Bias in Operator Learning** We demonstrate the spectral bias phenomena in the neural operator training in Figure 4.2. We compare HT-Net trained with $H^1$ and $L^2$ losses, and show the evolution of errors in the frequency domain as well as the loss curves. The comparison shows that HT-Net with $H^1$ loss has faster decay for high frequencies, and the errors over all the frequencies decay more uniformly. The HT-Net trained with $H^1$ loss also has better testing and generalization errors.

**Generalization Errors** HT-Net generalizes better on test data from samples out of distribution. In addition to what we have shown in (c,d) of Figure 4.2, we compare HT-Net with other models in Table 2. The models are trained with the rough two-phase dataset in Table 4.1 and tested on out-of-distribution data. FNO2D $H^1$ is FNO model but trained using our $H^1$ loss. HT-Net outperforms other models by order of magnitude.

| | FNO2D | FNO2D $H^1$ | MWT | HT-NET |
|---|---|---|---|---|
| $n = 256$ | 20.27 | 11.49 | 21.901 | **3.182** |

Table 2: Relative $L^2$ error ($\times 10^{-2}$) for out of distribution data, with $a_{\max} = 12$, $a_{\min} = 3$ and $c = 18$.

### 4.3 NAVIER-STOKES EQUATION

We consider the 2D Navier-Stokes equation in vorticity form on the unit torus, which is also benchmarked in Li et al. (2021)

$$\partial_t w(x,t) + u(x,t) \cdot \nabla w(x,t) = \nu \Delta w(x,t) + f(x), \quad x \in (0,1)^2, t \in (0,T]$$
$$\nabla \cdot u(x,t) = 0, \qquad\qquad x \in (0,1)^2, t \in [0,T]$$
$$w(x,0) = w_0(x), \qquad\qquad x \in (0,1)^2$$

with velocity $u$, vorticity $w = \nabla \times u$, initial vorticity $w_0$, viscosity $\nu > 0$ ($\sim \mathrm{Re}^{-1}$ with $\mathrm{Re}$ being the Reynolds number), and forcing term $f$. We learn the operator $\mathcal{S} : w(\cdot, 0 \leq t \leq 10) \to w(\cdot, 10 \leq t \leq T)$, mapping the vorticity up to time 10 to the vorticity up to some later time $T > 10$. We experiment with viscosities $\nu = 1e-3, 1e-4, 1e-5$, and decrease the final time $T$ accordingly as the Reynolds numbers increase and the dynamics become chaotic.

**Time dependent neural operator** Following the setup in Li et al. (2021), we fix the resolution as $64 \times 64$ for both training and testing. The ten time-slices of solutions $w(\cdot, t)$ at $t = 0, ..., 9$ are taken as the input data to the neural operator $\mathcal{N}$ which maps the solutions at the previous 10 timesteps to the next time step (2D functions to 2D functions). This procedure, often referred to as the rolled-out prediction, can be repeated recurrently until the final time $T$. We list the results of HT-Net, FNO-3D (convolution in space-time), FNO-2D, U-Net in Ronneberger et al. (2015) in Table 3, and HT-Net is significantly better than other methods.

|  | #Parameters | $T = 50$ $\nu = 1e-3$ $N = 1000$ | $T = 30$ $\nu = 1e-4$ $N = 1000$ | $T = 30$ $\nu = 1e-4$ $N = 10000$ | $T = 20$ $\nu = 1e-5$ $N = 1000$ |
|---|---|---|---|---|---|
| FNO-3D | $6,558,537$ | 0.0086 | 0.1918 | 0.0820 | 0.1893 |
| FNO-2D | 414,517 | 0.0128 | 0.1559 | 0.0834 | 0.1556 |
| U-Net | $24,950,491$ | 0.0245 | 0.2051 | 0.1190 | 0.1982 |
| HT-Net | $10,707,204$ | **0.0050** | **0.0517** | **0.0194** | **0.0690** |

Table 3: Benchmarks for the Navier Stokes equation. The resolution is $64 \times 64$ for both training and testing, and all models are trained for 500 epochs. $N$ is the size of the training samples.

## 5 CONCLUSION

We have built HT-Net, a hierarchical transformer based operator learning model for multiscale PDEs, which allows nested computation of features and self-attentions, and provides a hierarchical representation for the multiscale solution space. The reduce, multilevel local aggregation, and decompose operations form a fine-coarse-fine V-cycle for the feature update. We also introduced the empirical $H^1$ loss to reduce the spectral bias in the multiscale operator learning. HT-Net provides much better accuracy and robustness compared with state-of-the-art (SOTA) neural operators, which is demonstrated by several multiscale benchmarks.

Limitation and outlook (1) The current implementation of HT-Net relies on a regular grid. The extension to data clouds and graph neural networks will offer more opportunities to take advantage of the hierarchical representation. (2) The current implementation of the attention-based operator lacks the flexibility to generalize to a different resolution as FNO (Li et al., 2021). However, for multiscale problems, the discretization invariance of FNO and MWT models may be hampered by the aliasing error in the frequency domain, which is validated by the experiment results. Recent convergence analysis of operator learning methods such as Kovachki et al. (2021); Lanthaler et al. (2022) leverage the smoothness of the solutions, while multiscale PDEs usually have lower regularity and do not fall into such categories. For HT-Net, it might be possible to achieve (approximate) discretization invariance by using proper sampling and interpolation modules, we expect that a full treatment will be operator adaptive.

ETHICS STATEMENT

This work proposes a hierarchical transformer operator learning model for multiscale PDEs. As stated in the introduction, solving multiscale PDEs is associated with some of the most challenging practical problems, such as reservoir modeling, fracture and fatigue prediction, high frequency scattering, weather forecasting, etc. Potentially, HT-Net solvers could help to reduce the prohibitively expensive computational cost of those simulations. The negative consequences are not obvious. Though, in theory, any technique can be misused, it is not likely to happen at the current stage.

REPRODUCIBILITY STATEMENT

The datasets for Section 4 are either downloaded (for smooth two-phase coefficients and Navier-Stokes) from https://github.com/zongyi-li/fourier_neural_operator, or generated (for rough Darcy coefficients) using the code from the same website. We implemented $\mathcal{P}_1$ finite element method in MATLAB to solve equation 2.1 with multiscale trigonometric coefficients in Appendix F.1, and in FreeFEM to solve the Helmholtz equation in Appendix F.2. We have included introductions of the relevant mathematical and data generation concepts in the appendix.

We put the code for and also a link for datasets at the anonymous Github page https://github.com/Shengren-Kato/VFMM-ICLR2023.git. Supplementary descriptions of the code are also provided in the page.

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

## A  MULTISCALE PDE

In this section, we introduce some mathematical and numerical concepts related to multiscale PDEs.

### A.1   ASYMPTOTIC HOMOGENIZATION

We introduce some basic formulation of asymptotic homogenization, by assuming that $a(x) = a(\frac{x}{\varepsilon})$ with 1-periodic funciton $a(\cdot)$ in equation 2.1, namely,

$$
\begin{cases}
-\text{div}\left(a\left(\frac{x}{\varepsilon}\right)\nabla u^{\varepsilon}\right) = f, & x \in D, \subset \mathbb{R}^d, a(\cdot) \text{ periodic in } y = x/\varepsilon, \\
u^{\varepsilon} = 0, & x \in \partial D.
\end{cases}
$$

It can be derived from the asymptotic expansion of the two-scale function $u^{\varepsilon} = u(x, x/\varepsilon)$ in $\varepsilon$, and justified rigorously that $u^{\varepsilon} \rightharpoonup u_0$ in $H^1$, with $u_0(x)$ the solution of the homogenized problem

$$
\begin{cases}
-\text{div}_x(a^0 \nabla_x u_0) = f, & x \in D, \\
u_0 = 0, & x \in \partial D.
\end{cases}
$$

which contains only coarse scale information. The homogenized coefficient $a^0$ can be computed by the formula $a_{ij}^0 = \int_Y (e_i + \nabla \chi_i)^T a(y)(e_j + \nabla \chi_j) dy$, where $\chi_i$ solves the cell problems given by

$$
\begin{cases}
-\text{div}_y\left(a\left(\nabla_y \chi_i + e_i\right)\right) = 0, & x \in Y, 1 \le i \le d, \\
\chi_i \in H^1_{\text{per}}(Y).
\end{cases}
$$

with $Y$ being the $d$ dimensional torus. Asymptotic homogenization provides an approximate solution $\hat{u}^{\varepsilon} = u_0 - \varepsilon \sum \chi_i\left(\frac{x}{\varepsilon}\right)\frac{\partial u_0}{\partial x_i}$ such that $\|u^{\varepsilon} - \hat{u}^{\varepsilon}\|_{H^1} \le c\varepsilon^{\frac{1}{2}}$. We note that to find the homogenization approximation $\hat{u}^{\varepsilon}$ only requires the coarse-scale solution $u_0$ and precomputation of $d$ cell problems for $\chi_i$ which do not depend on $f$ and $D$.

### A.2   NUMERICAL HOMOGENIZATION AND MULTILEVEL/MULTIGRID METHODS

Given the smallest scale of the multiscale problem $\varepsilon$ and a coarse computational scale determined by the available computational power and the desired precision, with $\varepsilon \ll H \ll 1$. The goal of numerical homogenization is to construct a finite-dimensional approximation space $V_H$ and to seek an approximate solution $u_H \in V_H$, such that, the accuracy estimate $\|u - u_H\| \le CH^{\alpha}$ holds for optimal choices of the norm $\|\cdot\|$ and the exponent $\alpha$, and optimal computational cost holds with $V_H$ constructed via pre-computed subproblems which are localized, independent and do not depend on the RHS and boundary condition of the problem.

In recent two decades, great progress has been made in this area (Hou et al., 1999; Målqvist & Peterseim, 2014; Owhadi & Zhang, 2007), approximation spaces with optimal accuracy (in the sense of Kolmogorov $N$-width (Berlyand & Owhadi, 2010)) and cost can be constructed for elliptic equations with fixed rough coefficients. For multiscale PDEs, operator learning methods can be seen as a step forward from numerical homogenization, since they can be applied to an ensemble of coefficients, and the decoder can be interpreted as the basis of the underlying problem as well.

For multiscale PDEs, multilevel/multigrid methods can be seen as the multilevel generalization of the numerical homogenization methods. Numerical homogenization can provide coarse spaces with optimal approximation and localization properties. Recently, operator-adapted wavelets (gamblets) (Owhadi, 2017; Xie et al., 2019) have been developed, and enjoy three properties that are ideal for the construction of efficient direct methods: scale orthogonality, well-conditioned multi-resolution decomposition, and localization. Gamblets has been generalized to solve Navier-Stokes equation (Budninskiy et al., 2019) and Helmholtz equation (Hauck & Peterseim, 2022) efficiently.

## B   HIERARCHICAL MATRIX PERSPECTIVE

The hierarchically nested attention in Algorithm 1 resembles the celebrated hierarchical matrix method (Hackbusch, 2015), in particular, the $\mathscr{H}^2$ matrix from the perspective of matrix operations. In the following,

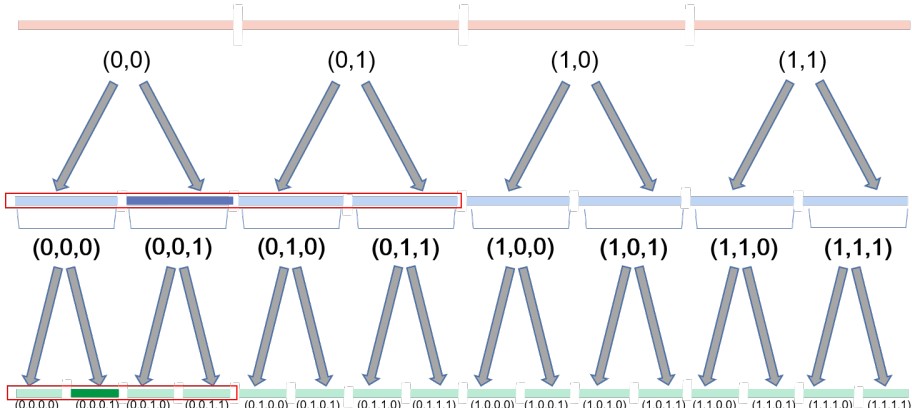

Figure B.1: Hierarchical discretization of 1D domain. The coarsest level partition is plotted as the top four segments in pink. The segment $(0,0)$ is further partitioned into two child segments $(0,0,0)$ and $(0,0,1)$. During the reducing process, the computation is performed from bottom to top to obtain coarser level tokens, for example, the $(0,0)$ tokens are obtained by learnable reduce operations $\mathcal{R}^{(2)}$ and $\mathcal{R}^{(1)}$ on tokens $(0,0,0)$ and $(0,0,1)$. When generating the high-resolution tokens/features, the computation is from up to bottom by learnable decomposition operations $\mathcal{D}^{(1)}$ and $\mathcal{D}^{(2)}$. The red frames show examples of attention windows at each level.

we take the binary tree-like hierarchical discretization shown in Fig B.1 as an example to illustrate the reduce operation, decompose operation, and multilevel token aggregation in Algorithm 1 in matrix formulas.

**STEP 0:** Given features $\boldsymbol{f}^{(r),t}$, compute the queries $\boldsymbol{q}_i^{(r),t}$, keys $\boldsymbol{k}_j^{(r),t}$, and values $\boldsymbol{v}_i^{(r),t}$ for $j \in \mathcal{I}^{(r)}$.

Starting from the finest level features $\boldsymbol{f}_i^{(r),t}, i \in \mathcal{I}^{(r)}$, the queries $\boldsymbol{q}^{(r),t}$ can be obtained by

$$
\begin{bmatrix} \vdots \\ \boldsymbol{q}_i^{(r),t} \\ \vdots \end{bmatrix} = \underbrace{\begin{bmatrix} \boldsymbol{W}^{Q,(r)} & & & \\ & \boldsymbol{W}^{Q,(r)} & & \\ & & \ddots & \\ & & & \boldsymbol{W}^{Q,(r)} \end{bmatrix}}_{|\mathcal{I}^{(r)}|} \left. \begin{bmatrix} \vdots \\ \boldsymbol{f}_i^{(r),t} \\ \vdots \end{bmatrix} \right\} |\mathcal{I}^{(r)}|,
$$

the keys $\boldsymbol{k}^{(r),t}$ and values $\boldsymbol{v}^{(r),t}$ can follow the similar procedure.

**STEP 1:** For $m = r - 1 : 1$, **Do** the reduce operations $\boldsymbol{q}_i^{(m),t} = \mathcal{R}^{(m)}(\{\boldsymbol{q}_j^{(m+1),t}\}_{j \in i^{(m,m+1)}})$ and also for $\boldsymbol{k}_i^{(m),t}$ and $\boldsymbol{v}_i^{(m),t}$, for any $i \in \mathcal{I}^{(m)}$.

If $\mathcal{R}^{(m)}$ is linear, the reduce operations correspond to $\begin{bmatrix} \vdots \\ \boldsymbol{q}_i^{(m),t} \\ \vdots \end{bmatrix} = \mathbf{R}^{(m),t} \begin{bmatrix} \vdots \\ \boldsymbol{q}_i^{(m+1),t} \\ \vdots \end{bmatrix}$ .The reduce

matrix is given by

$$\mathbf{R}^{(m)} := \left.\left[\begin{array}{ccccccc} R_0^{(m)} & R_1^{(m)} & & & & \\ & & R_0^{(m)} & R_1^{(m)} & & \\ & & & & \ddots & \ddots & \\ & & & & & R_0^{(m)} & R_1^{(m)} \end{array}\right]\right\}|\mathcal{I}^{(m)}|,$$

$$\underbrace{\qquad\qquad\qquad\qquad\qquad\qquad\qquad\qquad\qquad}_{|\mathcal{I}^{(m+1)}|}$$

and $\boldsymbol{R}_0^{(m)}, \boldsymbol{R}_1^{(m)} \in \mathbb{R}^{\mathcal{C}(m-1) \times \mathcal{C}(m)}$ are matrices parametrized by linear layers in our paper (in practice, queries, keys, and values use different $\boldsymbol{R}_0^{(m)}, \boldsymbol{R}_1^{(m)}$ to enhance the expressivity). In general, these operators $\mathcal{R}^{(m)}$ are not limited to linear operators. The composition of nonlinear activation functions would help increase the expressivity. The nested learnable operators $\mathcal{R}^{(m)}$ also induce the channel mixing and is equivalent to a structured parameterization of $\boldsymbol{W}^Q, \boldsymbol{W}^V, \boldsymbol{W}^K$ matrix for

the coarse level tokens, in the sense that, inductively, $\left[\begin{array}{c} \vdots \\ \boldsymbol{q}_i^{(m),t} \\ \vdots \end{array}\right] = \mathbf{R}^{(m)} \cdots \mathbf{R}^{(r-1)} \left[\begin{array}{c} \vdots \\ \boldsymbol{q}_i^{(r),t} \\ \vdots \end{array}\right] =$

$$\mathbf{R}^{(m)} \cdots \mathbf{R}^{(r-1)} \left[\begin{array}{cccc} \boldsymbol{W}^{Q,(r)} & & & \\ & \boldsymbol{W}^{Q,(r)} & & \\ & & \ddots & \\ & & & \boldsymbol{W}^{Q,(r)} \end{array}\right] \left[\begin{array}{c} \vdots \\ \boldsymbol{f}_i^{(r),t} \\ \vdots \end{array}\right].$$

**STEP 2:** With the $m$-th level queries and keys, we can calculate the local attention matrix $\boldsymbol{G}_{\text{loc}}^{(m),t}$ at $m$-th level with $G_{\text{loc,i,j}}^{(m),t} := \exp(\boldsymbol{q}_i^{(m),t} \cdot \boldsymbol{k}_j^{(m),t})$ for $i \in \mathcal{N}^{(m)}(j)$, or $i \sim j$.

**STEP 3:** The decompose operations, opposite to the reduce operations, correspond to the transpose of the following matrix in the linear case,

$$\mathbf{D}^{(m)} := \left.\left[\begin{array}{ccccccc} D_0^{(m)} & D_1^{(m)} & & & & \\ & & D_0^{(m)} & D_1^{(m)} & & \\ & & & & \ddots & \ddots & \\ & & & & & D_0^{(m)} & D_1^{(m)} \end{array}\right]\right\}|\mathcal{I}^{(m)}|,$$

$$\underbrace{\qquad\qquad\qquad\qquad\qquad\qquad\qquad\qquad\qquad}_{|\mathcal{I}^{(m+1)}|}$$

The $m$-th level aggregation in Figure 3.2 contributes to the final output $\boldsymbol{f}^{(r),t+1}$ in the form

$\mathbf{D}^{(r-1),\mathsf{T}} \cdots \mathbf{D}^{(m),\mathsf{T}} \boldsymbol{G}_{\text{loc}}^{(m)} \mathbf{R}^{(m)} \cdots \mathbf{R}^{(r-1)} \left[\begin{array}{c} \vdots \\ \boldsymbol{v}_i^{(r),t} \\ \vdots \end{array}\right]$. Eventually, aggregations at all $r$ levels in one V cycle

can be summed up as

$$\left[\begin{array}{c} \vdots \\ \boldsymbol{f}_i^{(r),t+1} \\ \vdots \end{array}\right] = \left(\sum_{m=1}^{r-1} (\mathbf{D}^{(r-1),\mathsf{T}} \cdots \mathbf{D}^{(m),\mathsf{T}} \boldsymbol{G}_{loc}^{(m)} \mathbf{R}^{(m)} \cdots \mathbf{R}^{(r-1)}) + \boldsymbol{G}_{\text{loc}}^{(r)}\right) \left[\begin{array}{c} \vdots \\ \boldsymbol{v}_i^{(r),t} \\ \vdots \end{array}\right]. \qquad \text{(B.1)}$$

The hierarchical attention matrix $\boldsymbol{G}_h := \sum_{m=1}^{r-1} (\mathbf{D}^{(r-1),\mathsf{T}} \cdots \mathbf{D}^{(m),\mathsf{T}} \boldsymbol{G}_{loc}^{(m)} \mathbf{R}^{(m)} \cdots \mathbf{R}^{(r-1)}) + \boldsymbol{G}_{\text{loc}}^{(r)}$ in equation B.1 resembles the three-level $\mathscr{H}^2$ matrix decomposition illustrated in the following Figure B.2, we

also refer to Hackbusch (2015) for a detailed description. The sparsity of the matrix lies in the fact that the attention matrix is only computed for pairs of tokens within the neighbor set. The $\mathscr{H}^2$ matrix-vector multiplication in B.1 also implies $\mathcal{O}(N)$ complexity of the Algorithm 1.

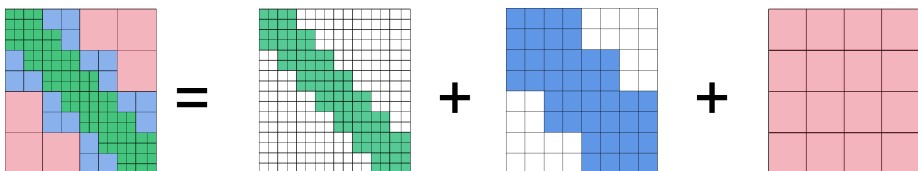

Figure B.2: A demonstration of the decomposition of attention matrix into three levels of local attention matrix.

Note that, the local attention matrix at level $\mathcal{I}^{(1)}$ (pink), level $\mathcal{I}^{(2)}$ (blue) and level $\mathcal{I}^{(3)}$ (green ) are $\boldsymbol{G}_{loc}^{(1)}$, $\boldsymbol{G}_{loc}^{(2)}$ and $\boldsymbol{G}_{loc}^{(3)}$, respectively. However, when considering their contributions to the finest level, they are equivalent to the attention matrix $\mathbf{D}^{(2),\mathsf{T}}\mathbf{D}^{(1),\mathsf{T}}\boldsymbol{G}_{loc}^{(1)}\mathbf{R}^{(1)}\mathbf{R}^{(2)} \in \mathbb{R}^{\mathcal{I}^{(3)}} \times \mathbb{R}^{\mathcal{I}^{(3)}}$ (pink), $\mathbf{D}^{(2),\mathsf{T}}\boldsymbol{G}_{loc}^{(2)}\mathbf{R}^{(2)} \in \mathbb{R}^{\mathcal{I}^{(3)}} \times \mathbb{R}^{\mathcal{I}^{(3)}}$ (blue) and $\boldsymbol{G}_{loc}^{(3)}$ (green), as demonstrated in Figure B.2. Each pink block and blue block are actually low-rank sub-matrix with rank $\mathcal{C}^{(1)}$ and rank $\mathcal{C}^{(2)}$, respectively, by definition.

## C    PROOF OF PROPOSITION 3.1

*Proof*    For each level $m$, the cost to compute equation 3.3 is $c(|\mathcal{I}^{(m)}|\mathcal{C}^{(m)})$ since for each $i \in \mathcal{I}^{(m)}$ the cardinality of the neighbour set $\mathcal{N}^{(m)}(i)$ is bounded by a constant $c$. The reduce operation $\boldsymbol{f}_i^{(k-1)} = \mathcal{R}^{(k-1)}(\{\boldsymbol{f}_j^{(m)}\}_{j\in i^{(k-1,k)}})$ costs at most $|\mathcal{I}^{(m)}|\mathcal{C}^{(m)}\mathcal{C}^{(k-1)}$ flops and so does the decompose operation at the same level. Therefore, for each level, the operation cost is $c(|\mathcal{I}^{(m)}|\mathcal{C}^{(m)}) + 2|\mathcal{I}^{(m)}|\mathcal{C}^{(m)}\mathcal{C}^{(m-1)}$. When $\mathcal{I}$ is a quadtree, $\mathcal{I}^{(r)} = N$, $\mathcal{I}^{(r-1)} = N/4, \cdots, \mathcal{I}^{(1)} = 4$, therefore the total computational cost $\sim \mathcal{O}(N)$.    $\square$

## D    IMPLEMENTATION DETAILS

In the HT-Net implementation, we follow the window attention scheme in Liu et al. (2021b) for the definition of the neighborhood $\mathcal{N}^{(\cdot)}(\cdot)$ in equation 3.2 and equation 3.3. In this paper, we choose $r = 3$ as the depth of the HT-Net, GeLU as the activation function, and a CNN-based patch embedding module to transfer the input data into features.

For a dataset with resolution $n_f \times n_f$, such as in the multiscale elliptic equation benchmark 4.2, the input feature $\boldsymbol{f}^{(3)}$ is represented as a tensor of size $n \times n \times C$ via patch embedding. The self-attention is first computed within a local window on level 3. Then the reduce layer concatenates the features of each group of $2 \times 2$ neighboring tokens and applies a linear transformation on the $4C$-dimensional concatenated features on $\frac{n}{2} \times \frac{n}{2}$ level 2 tokens, to obtain level 2 features $\boldsymbol{f}^{(2)}$ as a tensor of the size $\frac{n}{2} \times \frac{n}{2} \times 2C$. The procedure is repeated from level 2 to level 1 with $\boldsymbol{f}^{(1)}$ of size $\frac{n}{4} \times \frac{n}{4} \times 4C$.

We adopt the window attention scheme from Liu et al. (2021b). One can refer to the implementation details there. For the decompose process, starting at level 1, a linear layer is applied to transform the $4C$-dimensional features $\boldsymbol{f}^{(1)}$ into $8C$-dimensional features. Each level 1 token with $8C$-dimensional features is decomposed into four level 2 tokens with $2C$-dimensional features and added to the level 2 feature $\boldsymbol{f}^{(2)}$

with output size of $\frac{n}{2} \times \frac{n}{2} \times 2C$. The procedure is repeated from level 2 level to level 3 with the output $\boldsymbol{f}(3)$ of size $n \times n \times C$.

We use the Adam optimizer with learning rate $1\mathrm{e}{-}3$, weight decay $1\mathrm{e}{-}4$ and the 1-cycle schedule as in Cao (2021). We use batch size 8 for experiments in Sections 4.2 and 4.3, and batch size 4 for experiments in Appendices F.1 and F.2.

**Evaluation setup** For the Darcy rough case in Table 1, we use a train-val-test split of the dataset with sizes 1280, 112, and 112, respectively. Each run is carried out with different seeds. For the Darcy smooth and multiscale trigonometric cases, we use a train-val-test split of the dataset with sizes 1000, 100 and 100, respectively. For the Navier-Stokes experiment in Section 4.3, we have tried $N = 1000$ and $N = 10000$ training samples as shown in Table 3, and the number of testing samples is 100 and 1000 respectively. The baseline's implementations are based on their official implementation if provided publicly. The results are comparable with the ones reported in the references.

All experiments are run on a NVIDIA A100 GPU.

## E    DATA GENERATION FOR THE TWO-PHASE COEFFICIENT IN SECTION 4.2

The two-phase coefficients and solutions are generated according to https://github.com/zongyi-li/fourier_neural_operator/tree/master/data_generation, and used as operator learning benchmarks in Li et al. (2021); Gupta et al. (2021); Cao (2021). The coefficients $a(x)$ are generated according to $a \sim \mu := \psi_\# \mathcal{N}\left(0, (-\Delta + cI)^{-2}\right)$ with zero Neumann boundary conditions on the Laplacian. The mapping $\psi : \mathbb{R} \to \mathbb{R}$ takes the value $a_{\max}$ on the positive part of the real line and $a_{\min}$ on the negative part. The push-forward is defined in a pointwise manner. The forcing term is fixed as $f(x) \equiv 1$. Solutions $u$ are obtained by using a second-order finite difference scheme on a suitable grid. The parameters $a_{\max}$ and $a_{\min}$ can control the contrast of the coefficient. The parameter $c$ can control the roughness (oscillation) of the coefficient, and the coefficient with a larger $c$ has rougher two-phase interfaces, as shown in Figure 4.1.

## F    ADDITIONAL EXPERIMENTS

### F.1    MULTISCALE TRIGONOMETRIC COEFFICIENT

In this experiment, we consider equation 2.1 with multiscale trigonometric coefficient adapted from Owhadi (2017), such that $D = [-1, 1]^2$, $a(x) = \prod_{k=1}^{6} (1 + \frac{1}{2}\cos(a_k \pi(x_1 + x_2)))(1 + \frac{1}{2}\sin(a_k \pi(x_2 - 3x_1)))$, with $a_k = \mathrm{uniform}(2^{k-1}, 1.5 \times 2^{k-1})$, and fixed $f(x) \equiv 1$. The reference solutions are obtained using $\mathcal{P}_1$ FEM on a $1023 \times 1023$ grid. Datasets of lower resolution are sampled from the higher resolution dataset by linear interpolation. The experiment results for the multiscale trigonometric case with different resolutions are shown in Table 4. HT-Net obtains the best relative $L^2$ error compared to other neural operators at various resolutions at 600 epochs. Multiwavelet neural operator MWT has the second best performance as it also possesses a multiresolution structure, but it does not adapt to the PDE. It is not surprising that FNO, an excellent smoother which filters higher frequency modes, fails to capture the high-frequency oscillations of the solution. In contrast, our method has a better performance in this respect. See Figure F.1 for illustrations of the coefficient and comparison of the solutions at the slice $x = 0$.

| | epochs=300 | | | epochs=600 | | |
|---|---|---|---|---|---|---|
| | n=128 | n=256 | n=512 | n=128 | n=256 | n=512 |
| FNO | 1.996 | 1.842 | 1.817 | 2.017 | 1.820 | 1.806 |
| GT | 1.524 | 1.070 | 1.093 | 1.448 | 0.938 | 0.970 |
| MWT | 1.115 | 1.007 | 1.006 | 1.112 | 0.985 | 0.977 |
| SWIN | 1.768 | 2.378 | 4.513 | 1.579 | 2.216 | 4.365 |
| HT-net | **0.581** | **0.671** | **0.659** | **0.537** | **0.614** | **0.631** |

Table 4: Relative error ($\times 10^{-2}$) of the multiscale trigonometric example.

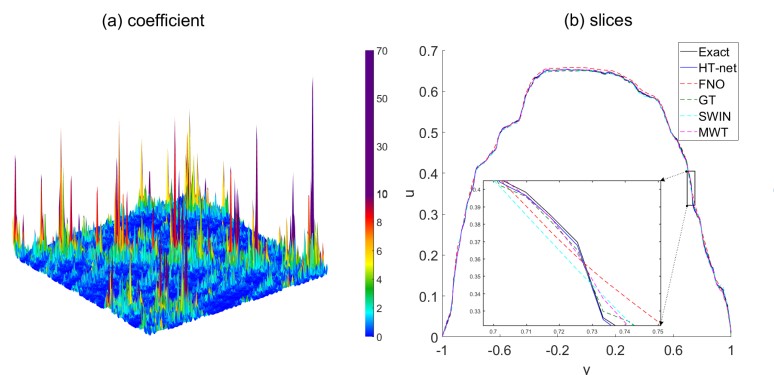

Figure F.1: (a) multiscale trigonometric coefficient, (b) slice of the solutions at $x = 0$.

## F.2 HELMHOLTZ EQUATION

We test the performance of HT-Net for the acoustic Helmholtz equation in highly heterogeneous media as an example of multiscale wave phenomena, whose solution is considerably expensive for complicated and large geological models. We adopt the setup from De Hoop et al. (2022), for the Helmholtz equation on the domain $D = [0, 1]^2$. Given frequency $\omega = 10^3$ and wavespeed field $c : \Omega \to \mathbb{R}$, the excitation field $u : \Omega \to \mathbb{R}$ solves the equation

$$
\begin{cases}
\left( -\Delta - \dfrac{\omega^2}{c^2(x)} \right) u = 0 & \text{in } \Omega, \\
\dfrac{\partial u}{\partial n} = 0 & \text{on } \partial\Omega_1, \partial\Omega_2, \partial\Omega_4, \\
\dfrac{\partial u}{\partial n} = 1 & \text{on } \partial\Omega_3,
\end{cases}
$$

where $\partial\Omega_3$ is the top side of the boundary, and $\partial\Omega_{1,2,4}$ are other sides. The wave speed field is $c(x) = 20 + \tanh(\tilde{c}(x))$, where $\tilde{c}$ is sampled from the Gaussian field $\tilde{c} \sim \mathcal{N}(0, \left( -\Delta + \tau^2 \right)^{-d})$, where $\tau = 3$ and $d = 2$ are chosen to control the roughness. The Helmholtz equation is solved on a $100 \times 100$ grid by finite element methods. We aim to learn the mapping from $c \in \mathbb{R}^{100 \times 100}$ to $u \in \mathbb{R}^{100 \times 100}$ as shown in Fig F.2.

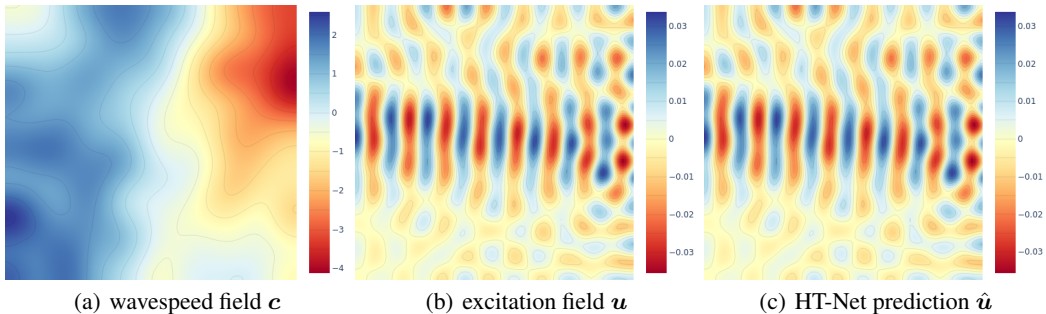

(a) wavespeed field $c$        (b) excitation field $u$        (c) HT-Net prediction $\hat{u}$

Figure F.2: The mapping $c \mapsto u$

The Helmholtz equation is notorious for solving. One reason is the so-called resonance phenomenon as the frequency $\omega$ is close to an eigenfrequency of the Helmholtz operator for some particular $c$. We found that it is necessary to use a large training dataset of size $8000$. The test dataset is of size $800$. All models are trained within $500$ epochs.

| Model | #Parameters | Evaluation time (ms) | $L^2$ relative error ($\times 10^{-2}$) |
|-------|-------------|----------------------|------------------------------------------|
| FNO | 3,282,068 | 5.8 | 2.301 |
| UNet | 17,261,825 | 0.1 | 42.90 |
| HT-Net | 47,632,003 | 3.0 | 0.687 |

Table 5: HT-Net outperforms the other models by a wide margin. The error of FNO is comparable with the reported results in De Hoop et al. (2022). We note that the four models benchmarked for the Helmholtz equation in De Hoop et al. (2022), including FNO and DeepONet, failed to reach an error less than $1 \times 10^{-2}$.

We also compare the evaluation time of the trained models in 5. Compared to HT-Net, FNO has both a larger error and a longer evaluation time. UNet, as a CNN based method, has much faster evaluation time (30 times faster than HT-Net) but the worst error (60 times as much as HT-Net).

### F.3 COMPARISON WITH CLASSICAL METHODS (FDM) AND MULTISCALE METHODS (GRPS)

We compare the accuracy and efficiency of HT-Net with two conventional solvers - finite difference method (FDM) as a typical classical method, and generalized rough polyharmonic splines (GRPS) (Liu et al., 2021a) as a typical multiscale method, for the Darcy rough benchmark. The results are listed in Table 6.

| Model | Evaluation time (s) | relative $L^2$ error ($\times 10^{-2}$) |
|-------|---------------------|------------------------------------------|
| FDM | 0.34 | 0.84 |
| GRPS | 18.9 | **0.02** |
| HT-Net | **0.003** | 0.58 |

Table 6: Relative $L^2$ error and evaluation time on the Darcy rough benchmark. We implement FDM and NH in MATLAB and measure the solution time on a CPU Intel(R) Core(TM) i7-10510U CPU @ 2.30 GHz. We measure the evaluation time of HT-Net on an NVIDIA A100 GPU. Compared with classical methods, HT-Net has comparable accuracy but needs much less time for evaluation. The reference solution is defined on grids $512 \times 512$, sampled from the test dataset. FDM solves the problem on a $256 \times 256$ grid. GRPS uses coarse bases of dimension $256^2$ to solve the same problem. HT-Net learns the solutioin operator on the same resolution.

## F.4  STUDY FOR HYPERPARAMETERS

| Model | $H^1$ relative error ($\times 10^{-2}$) | $L^2$ relative error ($\times 10^{-2}$) |
|---|---|---|
| HT-Net [3, 8, 80] | 1.843 | 0.648 |
| HT-Net [3, 8, 128] | 1.761 | 0.688 |
| HT-Net [3, 4, 80] | 1.898 | 0.710 |
| HT-Net [3, 4, 64] | 1.909 | 0.695 |
| HT-Net [3, 2, 80] | 2.030 | 0.707 |
| HT-Net [2, 8, 80] | 1.903 | 0.701 |

Table 7: Hyperparameter study.

We conduct studies for hyperparameters such as number of hierarchical levels (depth), window size, and feature dimension. We list the results for the Darcy rough benchmark in Table 7, where the notation HT-Net [3, 8, 80] represents HT-Net with 3 levels, window size 8, and feature dimension 80. The experiments are run for 100 epochs. The results show that larger values of hierarchical level, window size, and feature dimension might be helpful to reduce errors, though with a higher computational cost. To balance the model size, computational cost, and performance, we choose the hyperparameters as [3, 8, 80].

## F.5  MEMORY USAGE

| Model | Darcy smooth (res=211×211) | | Darcy rough (res=256×256) | |
|---|---|---|---|---|
| | Mem | CUDA Mem | Mem | CUDA Mem |
| FNO | 1.80 | 1.19 | 2.53 | 1.72 |
| GT | 2.85 | 3.85 | 5.17 | 11.64 |
| MWT | — | — | 2.54 | 8.36 |
| HT-NET | 4.17 | 7.09 | 4.98 | 8.91 |

Table 8: The memory usage of different models. The CUDA mem(GB) is the sum of $self\_cuda\_memory\_usuage$ from the PyTorch autograd profiler for 1 backpropagation. The mem(GB) is recorded from $nvidia - smi$ of the memory allocated for the active Python process during profiling.

We report the memory usage of different models for the Darcy smooth (with resolution 211×211) and Darcy rough (with resolution 256×256) benchmarks in Table F.5. The table shows that the memory usage of HT-Net is stable. For the resolution of 256×256, both MWT and GT consume larger or comparable CUDA memory compared with HT-Net, albeit HT-Net has much better accuracy.

