# OpenReview forum: "HT-Net: Hierarchical Transformer based  Operator Learning Model for Multiscale PDEs"
_ICLR.cc/2023/Conference — Submitted to ICLR 2023_

### Official Review · Reviewer_oKvB · 2022-10-21

**Confidence:** 4
**Correctness:** 3
**Technical Novelty And Significance:** 2
**Empirical Novelty And Significance:** Not applicable
**Recommendation:** 6

**Clarity, Quality, Novelty And Reproducibility:**

Overall the paper is mostly clear and probably reproducible, especially with the additional details provided in the appendices. As I mentioned previously, I have some reservations about the quality of the evaluation. With respect to the novelty of the approach, the main new idea appears to be the application of multiscale transformers to the domain of neural operators. Still, transformers have been considered for this application before (as the authors note), and the idea of multiscale transformers is also not new (see, e.g., Multiscale Vision Transformers by Fan et al. [2021]).

The readability of the paper could also be improved in some areas, and I provide a few minor comments below.
- The references are not typeset correctly; author names are given in plain text rather than being enclosed in parenthesis.
- Fig. 4.2: Use epochs on the x-axis for all plots for clarity (currently switches from y-axis to x-axis from the left to right panels).
- Fig. 4.2: The plots should compare the HT-Net trained with H1 loss vs. HT-Net trained with L2 loss on the same axes. This allows direct comparison of the performance of the network trained with the different loss functions. In that case, the results from evaluating the models with the L2 metric could be used for one plot and the other plot would show evaluating the models with the H1 metric.


**Strength And Weaknesses:**

The method appears to outperform the baseline approaches for learning neural operators by a fair margin, and so may be of interest to the large community working on physics-informed neural networks and neural operators.

I also find the appendix to be very helpful in supporting some of the arguments made in the main paper. For example, the elaboration on the connection to hierarchical matrix methods and additional implementation details.

The architecture and method are also fairly straightforward to understand.

On the flip side, since the main technical contribution of the paper seems to be the architecture, I was expecting to see more from the evaluation. For example, the following points.
- There are no qualitative comparisons to the baselines; the only qualitative results (showing the predicted solutions) in the main paper are two output solutions of the elliptic equation from the proposed method.
- While there is a discussion of computational complexity, it's not clear what the actual practical runtime and memory usage are. One of the main advantages of neural operators is that they usually offer significant improvements in runtime compared to conventional solvers (albeit at the cost of accuracy). In that respect, transformers are usually more computationally expensive than, e.g., CNN-based architectures, and so I think it's critical to include some analysis of the computational performance.
- The authors mention the "Galerkin Transformer"  (Cao 2021) as another transformer-based architecture introduced in the context of operator learning. I wonder why this was not evaluated? How does that method compare to the proposed architecture?
- There is no discussion/evaluation of some of the architectural choices. For example, how does the choice of window size affect the performance?

**Summary Of The Paper:**

The paper presents a multiscale method for operator learning based on a transformer architecture. Inspired by hierarchical methods for solving PDEs, the key idea is to add self-attention layers to a architecture with multiple downscaling and upscaling layers (i.e., reduce and decompose operations). The authors also use a Sobolev norm to train the network, which they find improves the recovery of high-frequency signal content. The method is evaluated by learning solution operators for multiple different differential equations, and appears to outperform other recent methods.

**Summary Of The Review:**

Overall, I lean slightly negative on this paper. The results appear to outperform the baselines, but I'm curious why the recent transformer architecture for neural operators was not included in the baselines, and there seem to be some missing evaluation items: analysis of runtime/memory, ablation study of architectural decisions (since the architecture is the main contribution), and qualitative comparison to baseline approaches. Perhaps the authors can address some of these concerns in their response, though.

---

> ### Author Response · Authors · 2022-11-18
> **Response to Reviewer 4 (oKvB)**
>
> We thank the reviewer for the valuable comments on our work, and we believe that addressing those concerns will help clarify the novelty and improve the quality of the manuscript.
>
> **Comparison with multiscale transformer models**
> Please see Common Concern 1 for the comparison of the proposed HT-Net model and existing multiscale transformers, which helps clarify the novelty of the HT-Net model.
>
> **Choice of Hyperparameters**
> We added studies for architectural hyperparameters, such as hierarchical level (depth), window size, and feature dimension in Appendix F.4. Please see Common Concern 2.
>
> **Readability**
> We have improved the exposition of the paper, in particular in the "Methods" section. Please see Common Concern 3.
>
> **Qualitative comparison**
> In addition to the predicted solutions in the original manuscript, we add plots of absolute errors of HT-Net solutions in Figure 4.1 (d) and FNO2D solutions in Figure 4.1 (e) for both Darcy smooth and Darcy rough benchmarks, which demonstrate the improved accuracy of HT-Net over FNO2D. We also updated the slice plot for the comparison of different models on the multiscale trigonometric benchmark in Figure F.1.
>
> **Computational performance**
>
> - We compare the accuracy and efficiency of HT-Net with two conventional solvers - finite difference method (FDM) as a typical classical method, and generalized rough polyharmonic splines (GRPS) as a typical multiscale method. The results are listed in Table 6, Appendix F.3. We see that the HT-Net evaluation is much faster than FDM (100 times faster) and NH (6000 times faster) with comparable degrees of freedom. On the other hand, HT-Net has better accuracy than FDM, and the HT-Net error is about 29 times as much as NH. Therefore, we see that HT-Net can serve as a fast forward solver with moderate accuracy compared with conventional solvers.
>
> - We also compare HT-Net with UNet, which has a CNN-based architecture; see Table 5 in Appendix F.2 for the Helmholtz equation benchmark. We see that UNet (0.1s) is faster than HT-Net (3.0s) by a factor of 30, but the error of UNet is more than 60 times as much as HT-Net.
>
> - We add a comparison of memory usage in Appendix F.5. We report the memory usage of different models for the Darcy smooth (with resolution 211 $\times$ 211) and Darcy rough (with resolution 256 $\times$ 256) benchmarks in Table 8. The table shows that the memory usage of HT-Net is stable. For the resolution of 256 $\times$ 256, both MWT and GT consume larger or comparable CUDA memory compared with HT-Net, albeit HT-Net has much better accuracy.
>
> **Comparison with transformer-based architectures**
> We have included the Galerkin transformer [Cao (2021)](https://arxiv.org/abs/2105.14995) with the name "GT Ln on K, V" as a baseline in Table 1 in the original manuscript. We renamed it "GT" in Table 1 in the revision. We also added the SWIN transformer with the name "SWIN" [Liu et al. (2021)](https://arxiv.org/abs/2103.14030) as a baseline for the existing multiscale transformer. HT-Net outperforms both existing transformer-based architectures by a wide margin.
>
>
> **Minor Comments**
> - We have redone the typeset of the references according to ICLR format. Namely, when the authors or the publication are included in the sentence, the citation is put in parenthesis using \citet{}; otherwise, the citation is put in parenthesis using \citep{}.
> - In Figure 4.2, the usage of epochs in (a,b) denotes the evolution of different frequency components, and the format (x-axis: frequency, y-axis: epochs) is consistent with Figure 1 and Figure 8 of [Rahaman, et al. (2018)](https://arxiv.org/abs/1806.08734) which started the study of spectral bias of neural networks. The usage of epochs in (c,d) is consistent with most machine learning literature, for example, Figure 4 and Figure 8(e) of [Rahaman, et al. (2018)](https://arxiv.org/abs/1806.08734), with the format (x-axis: epochs, y-axis: error/loss). Therefore, we prefer to keep the original format.
> - In Figure 4.2 (c,d), loss curves for (c) the HT-Net trained with $H^1$ loss and (d) the HT-Net trained with $L^2$ loss are shown. We now use the same scale in x-axis and the same log scale in y-axis in both figures (c,d). Therefore, it is rather straightforward to evaluate the performances of both models.

---

> > ### Comment · Reviewer_oKvB · 2022-11-23
> > **discussion**
> >
> > I looked through the other reviews and the author responses. I think the authors have done a good job of including additional results and running the requested baselines, and the comparison to other transformer architectures is very helpful. The method does seem to improve performance quite a bit (albeit with some increase in the amount of memory and parameters required as shown in the supplement).
> >
> > There are still a number of typos and grammatical errors in the revisions, but I think the authors can correct these before the final submission. At this point I'd be okay with accepting the paper, and I updated my rating.

---

> > > ### Author Response · Authors · 2022-11-24
> > > **Thanks for your advice**
> > >
> > > We thank the reviewer for raising the points and are happy to see that the concerns are addressed. We will certainly do careful proofreading to correct typos and grammatical errors for the final submission.

---

### Official Review · Reviewer_2aoA · 2022-10-24

**Confidence:** 4
**Clarity, Quality, Novelty And Reproducibility:** The paper is  clear enough, high qual…
**Correctness:** 4
**Technical Novelty And Significance:** 3
**Empirical Novelty And Significance:** 4
**Recommendation:** 10

**Strength And Weaknesses:**

The efficient hierarchical operator learning of the input-output mapping of parametric PDEs using attention. The emphasis on getting the high frequency components correctly is also required for performance.

**Summary Of The Paper:**

The manuscripts an attention based multi-grid solution for PDEs where different scales are present in the solution. This is done by a hierarchical attention structure that is used to map low feature solutions to high feature solutions and back. The hierarchy is created with a network architecture that resembles U-Net of Ladder architectures, spiced with transformer layers. The manuscript describes a hierarchical discretisation that have features at multiple scales. Reduce operators go to coarser level and decompose operation does the opposite.

A good choice as these have been shown to be able to handle multiple resolutions in an efficient manner using the transformer architecture.   The result are very encouraging and provide significant improvement over the state-of the -art.

**Summary Of The Review:**

The attention seems to be last world in many high performance application.  I think that in this case really for being able to bring the prior art features to the fine grid solution  in the decompose operator.  It would be nice to know (or have discussion) what are the salient features that are in the coarse solution that enable a successful generation of the high resolution solution. it is definitely coming as a prior from the training data.  All in all an excellent paper definitely of interest to people finding methods to solve PDEs in efficient way

---

> ### Author Response · Authors · 2022-11-18
> **Response to Reviewer 3 (2aoA)**
>
> We thank the reviewer for appreciating our results and the kind words! It is encouraging to see that the reviewer has found our work "strong"!
>
> In the following, we give some discussion on why the HT-model may enable a successful generation of high-resolution solutions.
>
> The generation of high-resolution solutions from coarse features or coarse measurements is a long-standing problem in the multiscale modeling and simulation community. For the multiscale PDEs with fixed parameters, important progress has been made from the perspective of numerical homogenization ([Engquist, Souganidis (2008)](https://www.cambridge.org/core/journals/acta-numerica/article/asymptotic-and-numerical-homogenization/F0BFD0BAF34C6AB468A7C9ECAB237715)), $\mathcal{H}$ matrix ([Hackbusch, (2015)](https://link.springer.com/book/10.1007/978-3-662-47324-5)) etc. Roughly speaking, the optimal basis set can be determined by variational formulation ([Hugues et al. (1998)](https://www.sciencedirect.com/science/article/pii/S0045782598000796)), singular value decomposition ([Babuska et al. (2011)](https://doi.org/10.1137/100791051)), or Bayesian inference ([Owhadi, (2014)](https://arxiv.org/abs/1406.6668)), and encoded in the reduce and decompose operation ([Owhadi, (2017)](https://doi.org/10.1137/15M1013894)). We believe that this should also be the case for the operator learning tasks.
>
> **Relation with vanilla attention** In the following, we give a brief and heuristic discussion on the relationship between vanilla attention matrix $\mathbf{G}$ and hierarchically nested attention matrix $\mathbf{G}_h$, and the roles of reduce and decompose operations. We conjecture that the off-diagonal blocks in $\mathbf{G}_h$ can serve as low-rank approximations of the off-diagonal blocks of the vanilla attention matrix $\mathbf{G}$. In [Rahimi, (2007)](https://papers.nips.cc/paper/3182-random-features-for-large-scale-kernel-machines), a random feature model was proposed to give a low-rank approximation to the Gaussian kernel, with sine and cosine transformations of features. This property was employed to construct efficient attentions, such as Performers  ([Choromanski et al. 2021](https://arxiv.org/abs/2009.14794)) and random feature attention ([Peng et al. 2021](https://arxiv.org/abs/2103.02143)). It might be possible to achieve low-rank approximation to the vanilla attention $\mathbf{G}$ at the off-diagonal blocks on hierarchical levels, by composing a particular nonlinear transform and constructing parameterized matrices $\mathbf{W}$ which act on features in reduce and decomposition operations. The hierarchical attention mechanism simply provides an efficient implementation of the approximation with linear computational cost. We plan to investigate this approximation property theoretically in our future work.

---

### Official Review · Reviewer_uEBd · 2022-10-25

**Confidence:** 3
**Correctness:** 4
**Technical Novelty And Significance:** 3
**Empirical Novelty And Significance:** 3
**Recommendation:** 8

**Clarity, Quality, Novelty And Reproducibility:**

The exposition of the paper can be improved as suggested above. The work is of high quality in terms of methodology and rigor, and I believe it is novel. Reproducibility is probably highly unlikely without the code.


**Strength And Weaknesses:**

The main strength of the paper is the novelty of the approach. To the best of my knowledge, this work is first to suggest a hierarchical neural method based on the hierarchical matrix approach. The machinery involved in devising such an approach is based on a large body of theory and practice. The application domain of multiscale PDE is challenging, and the evaluation is convincing.

The main weakness of the paper is its exposition. This is unfortunately a hard read, which probably limit the impact of the paper outside the sub-community of PDE and numerical methods savvies. On the one hand, compressing the necessary details to nine pages is definitely a challenging task. On the other hand, illustrations of key operations would definitely help the reader. For instance, the discussion related to hierarchical discretization could benefit from an illustration showing a specific case. A similar comment can be made regarding the reduce and decomposition actions. I would also consider to de-clutter the current notations. Perhaps instead of using index notations, you can switch the operators performing reduce/decomposition.

Another potential weakness is the evaluation setup. How do you choose hyper-parameters for the network? Is there a validation set? How many times every experiment is run? Are the results reported in e.g., Tab. 1 for the basline approaches reported elsewhere (say in Li et al. 2021) or these are new numerics you achieved? Essentially, all these questions aim toward understanding whether there is a mature benchmark for this problem domain.

Finally, briefly discussing the limitations of the approach is needed. For instance, is it possible to use the method (probably not as is) for unstructured domains such as point clouds or general domains where natural neighborhood information is not avilable?


**Summary Of The Paper:**

This paper introduces a new hierarchical method to solve multiscale PDE problems of structured domains using attention-based temporal updates and hierarchical matrix reduce and decomposition. The method is evaluated on standard benchmark 2D multiscale PDE problems, showing promising results in comparison to existing work.


**Summary Of The Review:**

The task considered in this paper of solving multiscale PDE problems is highly challenging and complex. The proposed method tackles the problem in a novel approach, suggesting to compute hierarchical features in a v-shape updates procedure, going over all different hierarchical levels. As mentiond above, this is unfortunately a tough read which may potentially limit the impact of the approach and its wider accpetance. This in itself does not mean the paper should not be published, but rather I encourage the authors to make the effort to improve their exposition.

---

> ### Author Response · Authors · 2022-11-18
> **Response to Reviewer 2 (uEBd)**
>
> We thank the reviewer for acknowledging the novelty of our results and for the constructive suggestions.
>
> **Experiments**
> We address the choice of hyperparameters, validation set, and other experimental details in Common Concern 2.
>
> **Exposition**
> We have improved the exposition of the methods.
> - We illustrate the quadtree hierarchical discretization of the unit square in Figure 3.1.
> - We introduce the matrix-based formula of the linear reduce and decompose operations in the revision, for the specific case of quadtree hierarchical discretization.
> - We have rewritten Appendix B, which is consistent with the matrix-based notations in the main text. We also explained the calculation of hierarchical attention from the hierarchical matrix perspective in Appendix B.
>
> Please also see more details in Common Concern 3.
>
> **Limitations of the Approach**
> We add a paragraph to address the limitation of the current approach, as well as possible future directions in the Conclusion section of the revision. In particular, to perform the hierarchical attention in the HT-Net model for unstructured data such as point clouds or general domains, we do need neighborhood information, and a hierarchical clustering algorithm such as those implemented in algebraic multigrid methods (AMG) is the key for its efficient implementation. Another possibility is to use the graph neural network model such as [Li et al. (2020)](https://arxiv.org/abs/2003.03485).

---

### Official Review · Reviewer_Pg7t · 2022-11-03

**Confidence:** 3
**Correctness:** 2
**Technical Novelty And Significance:** 2
**Empirical Novelty And Significance:** 3
**Recommendation:** 5

**Clarity, Quality, Novelty And Reproducibility:**

* The clarity of the paper is okay for the most part, except for the "Hierarchical Discretization" paragraph which is particularly hard to follow with complicated notations associated to the indexes.

* There are quite a bit of typos scattered throughout the paper.

* I have appreciated that the authors have provided the code.

**Strength And Weaknesses:**

I have mixed feelings about this paper. I have appreciated the connection made with hierarchical matrices. The experimental results are also interesting, assuming the experiments have been conducted correctly (see comments below).


However, relation with previous work seems to have treated very poorly, which makes the novelty of the paper hard to evaluate.

For instance, the sentence "The idea is reflected in Liu et al. (2021); Zhang et al. (2022) to a certain degree." is vague and differences should be motivated and made much more precise, given most of the code is adapted from Liu et al. (2021).

For the second contribution (H1 loss), there is no mention to previous work, even though Sobolev type loss have already been proposed to train neural networks, e.g. in [Yu].

For the experiments section: Testing on a common number of epochs is not a good practice: the number of epochs should not be the same for each model, but chosen using the validation set.

As a more adapted baseline, have you tried FNO, preserving more modes in the Fourier transform in order to learn high-frequency outputs?

Lastly, it is not clear for me why this architecture would correspond to a Neural Operator. Can the network take as input arbitrary data (do you take as input positional encodings?). Does it satisfy the condition of "discretization invariance" [Kovachki]? I would like a more careful discussion on this point.


[Yu]: https://arxiv.org/pdf/2205.14300.pdf

[Kovachki]: https://arxiv.org/pdf/2108.08481.pdf

**Summary Of The Paper:**

The paper uses a hierarchical attention based neural networ to learn the solution operator associated to multiscale PDEs. In addition, the second contribution is the sobolev type norm used as the loss function, giving more weight to the higher frequencies of the target.

**Summary Of The Review:**

In the paper's current state, I would lean towards recommending rejection. However, I am willing to reconsider.

---

> ### Author Response · Authors · 2022-11-18
> **Response to Reviewer 1 (Pg7t) - Part 2**
>
> **Why HT-architecture corresponds to a neural operator**
> Attention-based operator is first proposed in [Cao (2021)](https://arxiv.org/abs/2105.14995), namely the GT method, where attentions are interpreted as Petrov–Galerkin-type projection in the infinite-dimensional function spaces. In that sense, HT-Net can be interpreted as a hierarchical (operator-adapted) Galerkin or Petrov-Galerkin approximation of the solution space. Also, Cao (2021) justifies empirically that GT has discretization invariance in the viscous Burgers benchmark (Table 2a in Cao (2021), also see the rebuttal of Cao (2021) at https://openreview.net/forum?id=ssohLcmn4-r), which has smooth solutions.
>
> **On arbitrary input data**
> In our current implementation, the data are sampled on a regular grid. For arbitrary input data, the hierarchical discretization and the hierarchical attention mechanism can still work provided an efficient hierarchical clustering algorithm for the data points is available. Another possibility is to take advantage of the graph neural network architecture as in [Li et al. (2020)](https://arxiv.org/abs/2003.03485). We agree that those are promising future directions, though it is beyond the scope of the current paper.
>
> **Discretization Invariance**
> We want to address the question of discretization invariance from different perspectives.
> - In our opinion, the (approximate) discretization invariance or zero-shot super-resolution prediction are nice concepts, but they require that the output of an operator that processes a continuous signal by sampling it at a set of discrete points should be largely independent of the sample points. Although Fourier layers or wavelet layers are discretization-invariant, the aliasing errors (components of a signal which are above the Nyquist frequency) limit their accuracy and practical usage for multiscale problems.
> - Our experiment results show that HT-Net outperforms FNO2d with 12, 48, and 96 modes and MWT (multiwavelet) methods by a huge margin. We believe that for multiscale PDEs, a discretization invariant operator learning method is possible. Though the transfer mapping has to be operator adapted. For a multiscale PDE with fixed parameters, such transfer mapping has been extensively studied by the multiscale computing community, see our introduction in Appendix A.2 and references therein. That also partially explains why HT-Net, as an operator-adapted method, outperforms Fourier and Multiwavelet based methods.
> - This can also be reflected in recent theoretical results for operator learning such as [Lanthaler et al. (2022)](https://arxiv.org/abs/2102.09618v1); [Kovachki et al. (2021)](https://arxiv.org/abs/2107.07562v1), which leverage the smoothness of the solutions to prove convergence of FNO/DeepONet, etc. However, multiscale PDEs usually have lower regularity and do not fall into such categories.

---

> ### Author Response · Authors · 2022-11-18
> **Response to Reviewer 1 (Pg7t) - Part 1**
>
> We thank the reviewer for providing valuable feedback. We now take the opportunity to clarify the concerns raised by the reviewer.
>
> **Comparision with multiscale transformers**
> We have addressed this issue in Common Concern 1.
>
> **Validation Set**
> We have addressed this issue in Common Concern 2.
>
> **Exposition**
> We have revised the paragraph on "Hierarchical Discretization". Please see Common Concern 3.
>
>
> **H1 loss**
> The choice of loss functions is crucial for efficient training and robust generalization of neural network models. Due to the multiscale nature of the problem considered in this paper, we choose $H^1$ loss instead of the usual $L^2$ loss function, as it puts more "weights" on the high-frequency components.
>
> [Yu et al. (2022)](https://arxiv.org/pdf/2205.14300.pdf) adopted Sobolev norm based loss function in the context of function approximation, where $H^1$ loss is defined for the neural network function $f(\mathbf{x};\theta): \mathbf{R}^d \rightarrow \mathbf{R}$ itself, imposing an inductive spectral bias. In this study, $\mathbf{x}$ is the input, and $f$ is the output.
>
> On the contrary, our $H^1$ loss is defined on the target solution space, which measures the distance between the predicted solution $\hat{\mathbf{u}}:=\mathcal{N}(\mathbf{a})$ and the ground truth $\mathbf{u}$. The $H^1$ loss allows us to learn fine-scale features in the solution space. By comparison, $\mathbf{a}$ is the input, and $\mathbf{u}$ is the output, and the $H^1$-norm is taken with respect to the spatial variable $\mathbf{x}$, not the input $\mathbf{a}$. Therefore, the analysis in Yu et al. (2022) does not apply directly. To our best knowledge, using the Sobolev norm based loss function has never been well studied either empirically or theoretically in the context of operator learning.
>
> We present strong empirical evidence (shown in Figure 4.2 and Table 2) that the $H^1$ loss function leads to better generalization error both for in-distribution and out-distribution samples compared to the $L^2$ loss function. It will be interesting to investigate this question theoretically.
>
> **FNO with more modes**
>
> We add the experiment results for FNO2D (by default, with 12 modes) with more modes (48 and 96) in Table 1. For the Darcy smooth case, the errors do not improve much. For the Darcy rough case, the errors improved by a factor of 2 with 48 modes, but get saturated with 96 modes. For the multiscale trigonometric case, the errors do not improve for either 48 or 96 modes. In all those benchmarks, HT-Net outperforms FNO2D with a factor of 2-2.5 for $L^2$ errors, and a factor of 3-4 for $H^1$ errors.
>
> The results indicate that simply increasing the number of modes does not help much to learn multiscale solutions, in other words, they demonstrate the limitation of Fourier transform based method for multiscale problems. This is also related to the following discussion on "discretization invariance".

---

### Author Response · Authors · 2022-11-18
**Common Concern 3: Exposition of the method (from reviewers Pg7t, uEBd, oKvB)**

We revise the exposition of the method, in particular, the hierarchically nested attention by connecting the index-based notation with matrix-based notation, and also by adding more graphic illustrations.

- We illustrate the (quadtree) hierarchical discretization of a 2D unit square in Figure 3.1.
- We have rewritten the reduce and decompose operations. In particular, we have focused our presentation on the quadtree hierarchical discretization in Figure 3.1 and made those operations more concrete by giving their matrix-based formulas in the linear case, such that the reduce operation is of the form,
$$\mathbf{f}\_i^{(m),t} = \mathbf{R}\_0^{(m)}\mathbf{f}\_{(i,0)}^{(m+1),t}+\mathbf{R}\_1^{(m)}\mathbf{f}\_{(i,1)}^{(m+1),t}+\mathbf{R}\_2^{(m)}\mathbf{f}\_{(i,2)}^{(m+1),t}+\mathbf{R}\_3^{(m)}\mathbf{f}\_{(i,3)}^{(m+1),t},$$
where $\mathbf{R}\_{0}^{(m)}, \mathbf{R}\_{1}^{(m)}, \mathbf{R}\_{2}^{(m)}, \mathbf{R}\_{3}^{(m)} \in \mathbb{R}^{\mathcal{C}^{(m-1)}\times \mathcal{C}^{(m)}}$, and $(i,0)$, $(i,1)$, $(i,2)$, $(i,3)$ are the $(m+1)$-th level child nodes of $i\in \mathcal{I}^{(m)}$. Similarly, the decompose operation is of the form, for $s=0,1,2,3$,
$$\mathbf{f}\_{(i,s)}^{(m+1),t+\frac12} = \mathbf{D}\_s^{(m),T}\mathbf{f}\_i^{(m),t}.$$
Those matrix notations are also consistent with the 1D explanation in Appendix B.

- The exposition in the main text is further supplemented by the 1D explanation from the hierarchical matrix perspective in Appendix B. The global attention matrix is decomposed and calculated in three levels of  local attention matrices, as illustrated in Figure B.1 and Figure B.2.

---

### Author Response · Authors · 2022-11-18
**Common Concern 2: Experiment (from reviewers Pg7t, uEBd, oKvB)**

We have done the following work to improve the rigor and reproducibility of experiments.

**Validation**

In the original version of our manuscript, we choose a common number of epochs of 100 or 500 simply to follow baselines in the references, for example, [Li et al.](https://openreview.net/pdf?id=dh_MkX0QfrK), [Cao](https://openreview.net/forum?id=ssohLcmn4-r), [de Hoop et al.](https://arxiv.org/abs/2203.13181), and [Takamoto et al.](https://openreview.net/pdf?id=dh_MkX0QfrK). We follow the reviewers' suggestion in the revision and redo all the experiments in Table 1 using the train-validate-test scheme. For example, for the Darcy rough benchmark in Table 1, we use 1280 samples for training, 112 samples for validation, and 112 samples for testing; for the Darcy smooth and multiscale trigonometric benchmarks, we use 1000 samples for training, 100 samples for validation, and 100 samples for testing.


**Choice of Hyperparameters**

We conduct studies for hyperparameters such as the number of hierarchical levels (depth), window size, and feature dimension, and list the results for the Darcy rough benchmark in Table 7 in Appendix F.4.

**Other details**
- The Darcy smooth, Navier Stokes, and Helmholtz datasets are also benchmarked in [Li et al.](https://openreview.net/pdf?id=dh_MkX0QfrK), [Cao](https://openreview.net/forum?id=ssohLcmn4-r), [de Hoop et al.](https://arxiv.org/abs/2203.13181), and [Takamoto et al.](https://openreview.net/pdf?id=dh_MkX0QfrK), etc. The results for existing methods are consistent with references, and in most cases, the errors reported here are better than those in references, for example, by using the latest GitHub codes.

- We run each experiment 3 times with different seeds for the Darcy rough case in Table 1, and the standard deviations of the reported errors are small. This is also observed in [Gupta et al.](https://openreview.net/forum?id=LZDiWaC9CGL).
- We compare the accuracy and efficiency of HT-Net with two conventional solvers - finite difference method (FDM) as a typical classical method, and generalized rough polyharmonic splines (GRPS) as a typical multiscale method. The results are listed in Table 6, Appendix F.3.
- The performance of HT-Net on multiscale trigonometric and Helmholtz benchmarks are further improved; see Table 1 and Table 5.
- We also add qualitative comparisons for Dary smooth/rough (Figure 4.1(d,e)) and multiscale trigonometric (Figure F.1) benchmarks, which demonstrate the improved accuracy of HT-Net over other models.

---

### Author Response · Authors · 2022-11-18
**Common Concern 1: Comparison with existing multiscale/hierarchical transformers (from reviewers Pg7t, oKvB)**

We first emphasize the novelty of our HT-Net model and elucidate the essential differences with existing multiscale vision transformers.

In existing multiscale vision transformers such as [Liu et al. (2021)](https://arxiv.org/abs/2103.14030), [Fan et al. (2021)](https://arxiv.org/abs/2104.11227), and [Zhang et al. (2021)](https://arxiv.org/abs/2105.12723), spatial resolutions are sequentially reduced, and attentions are performed at each level separately, such that
- attentions are only used for a fixed level, and as such, there is no multilevel attention-based aggregation.
- in Liu et al. (2021) and Fan et al. (2021), only fine-coarse operations are performed such that coarse features can be used for classification and recognition tasks.
- in Zhang et al. (2021), a simple transpose of the fine-coarse operation is proposed for generation (coarse-fine) tasks. However, the fine-scale information might get lost since such a fine-coarse-fine cycle is done only sequentially, which is not preferable in the context of multiscale operator learning since fine-scale features are indispensable in predicting the solution of multiscale PDEs.

In the HT-Net architecture, we developed the hierarchically nested attention which extracts multiscale features for operator learning, motivated by multiscale numerical methods such as numerical homogenization and hierarchical matrix method.
- attention-based local aggregations are constructed at each level, and features from all levels are summed up to form the updated fine-scale features.
- the reduce/local aggregation/decompose operations mimic the hierarchical matrix method, in particular the $\mathcal{H}^2$ matrix, which enables the recovery of the fine details with linear cost.
- during the nested computation of features at all levels, we simultaneously parameterize the learnable matrices $W_Q, W_K, W_V$ in a nested way; see Appendix B for details from the hierarchical matrix perspective. This framework also allows us to compute the attentions among features/tokens across different levels, which will be investigated in the future.

As our focus is on the attention mechanism in the transformer architecture, in the implementation of HT-Net, we adopt the architecture (patch embedding, normalization, and window partition) of the SWIN transformer in Liu et al. (2021) but replace their attention with our hierarchical attention. To compare the empirical performances, in the revision, we add the original SWIN transformer in Liu et al. (2021) directly as a baseline (named SWIN, replacing the hierarchical attention mechanism of HT-Net with the attention mechanism in the original SWIN) of the multilevel vision transformer in Table 1.

---

### Author Response · Authors · 2022-11-18
**Overall response to all reviewers:**

We thank all the reviewers for their careful reading and thoughtful comments/suggestions. It is encouraging to see positive comments on the novelty and practical potential of our work. Also, the reviewers' feedbacks and suggestions have led us to further improve the clarity of the manuscript by adding more discussions and technical details.

We will first address the common concerns and then respond to each reviewer separately with further details. All the major revisions are marked blue in the revision.

---

### Decision · Program_Chairs · 2023-01-20

**Decision:**

Reject

**Justification For Why Not Higher Score:**

Presentation difficult to follow that could be improved. Difficult to assess on the real benefit of the proposed ideas in light of the experiments.


**Justification For Why Not Lower Score:**

N/A

**Metareview: Summary, Strengths And Weaknesses:**

The paper introduces a new method for solving multiscale PDEs based on transformer modules organized hierarchically according to a U-Net style architecture. The formulation is inspired from the theory of hierarchical matrices. The main novelty w.r.t. prior work exploiting U-Nets, hierarchical transformers or multiscale NNs based on hierarchical matrices is the introduction of a nested attention mechanism that allows for the aggregation of information at the different levels of the architecture. The model is evaluated on a multiscale elliptic equation and on a Navier-Stokes incompressible equation and is shown to outperform operator and U-Net like baselines.

The main originality of the paper lies in the aggregated attention mechanism that allows training, in a coordinated way, attention at the different levels when previous work consider independent attention for each layer in the hierarchy. The experiments show that the proposed mechanism allows the method to outperform Fourier Neural Operator (FNO) and U-Net like models. Initial critics pointed to a presentation that was difficult to follow and questioned the soundness of the experiments. The authors largely improved the presentation and added some experiments, this was appreciated by the reviewers. In my opinion the description of the method remains however hard to follow, and could probably be greatly simplified. This limits the paper impact for the ML community. The description and analysis of the experiments is incomplete, the lack of detail makes it difficult to understand exactly what was done and to assess the relevance of the conclusions of this experimental comparison. Although the authors made a significant effort for augmenting their comparisons during the rebuttal period, they did not use SOTA models. FNO models used here are a convenient baseline but are known to often underperform U-Net style architectures. There has been a series of U-shaped hierarchical models incorporating attention that have been developed, not only in vision but also for solving PDEs, that have been shown to outperform the U-Net baseline used here sometimes by an order of magnitude. Even U-shaped FNO are straightforward extensions of the initial FNO that have been recently proposed.

Here are some examples of references on modern U-Shaped architectures that the authors might want to consider for their evaluation.

- Ma et al. 2021 https://arxiv.org/pdf/2106.09301.pdf
- Chen et al. 2021 https://arxiv.org/pdf/2109.02183.pdf
- Rahman et al. 2022 https://arxiv.org/pdf/2204.11127.pdf (added here for information but too recent for being considered for the evaluation)

 Although the experimental comparison is quite decent, it is then difficult to conclude on the improvement brought of the proposed idea.


**Summary Of Ac-Reviewer Meeting:**

We had a discussion with reviewers oKvB and uEBo, and another one with reviewer Pg7t.
Reviewers oKvB and uEBo agree with the defaults highlighted above (tough reading and limited experiments) but conclude that after the new additions the paper could be accepted. Reviewer Pg7t agrees that the authors improved the exposition, but remains more mitigated. I re-read the paper carefully and I think that both the technical description and the experimental description and analysis are not satisfying.